

# Satellite quantification of methane emissions from South American countries: A high-resolution inversion of TROPOMI and GOSAT observations

Sarah E. Hancock[1], Daniel J. Jacob[1], Zichong Chen[1], Hannah Nesser[2], Aaron Davitt[3,4], Daniel J. Varon[1], Melissa P. Sulprizio[1], Nicholas Balasus[1], Lucas A. Estrada[1], James D. East[1], Elise Penn[1], Cynthia A. Randles[5], John Worden[2], Ilse Aben[6], Robert J. Parker[7,8], Joannes D. Maasakkers[6]

[1]Harvard University, School of Engineering and Applied Sciences, Cambridge, MA, 02138, USA
[2]Jet Propulsion Laboratory, California Institute of Technology, Pasadena, CA, 91011, USA
[3]WattTime, Oakland, CA, 94612, USA
[4]Climate TRACE, Denver, CO, 80022, USA
[5]United Nations Environment Program International Methane Emissions Observatory, Paris, France
[6]SRON Netherlands Institute for Space Research, Leiden, the Netherlands
[7]National Centre for Earth Observation, University of Leicester, Leicester, UK
[8]Earth Observation Science, School of Physics and Astronomy, University of Leicester, Leicester, UK

*Correspondence to*: Sarah E. Hancock (sarahhancock@g.harvard.edu)

**Abstract.** We use 2021 TROPOMI and GOSAT satellite observations of atmospheric methane in an analytical inversion to quantify national methane emissions from South America at up to 25 km × 25 km resolution. From the inversion, we derive optimal posterior estimates of methane emissions correcting the national anthropogenic emission inventories reported by individual countries to the United Nations Framework Convention on Climate Change (UNFCCC) and taken here as prior estimates. We also evaluate two alternative wetland emission inventories (WetCHARTs and LPJ-wsl) as prior estimates. Our best posterior estimates for wetland emissions are consistent with previous inventories for the Amazon but lower for the Pantanal and higher for the Parana. Our best posterior estimate of South American anthropogenic emissions is 48 (41-56) Tg a$^{-1}$, where numbers in parentheses are the range from our inversion ensemble. This is 55% higher than UNFCCC reports and is dominated by livestock (65% of anthropogenic total). We find that TROPOMI and GOSAT observations can effectively optimize and separate national emissions by sector for 10 of the 13 countries and territories in the region, 7 of which account for 93% of continental anthropogenic emissions: Brazil (19 (16-23) Tg a$^{-1}$), Argentina (9.2 (7.9-11) Tg a$^{-1}$), Venezuela (7.0 (5.5-9.9) Tg a$^{-1}$), Colombia (5.0 (4.4-6.7) Tg a$^{-1}$), Peru (2.4 (1.6-3.9) Tg a$^{-1}$), Bolivia (0.96 (0.66-1.2) Tg a$^{-1}$), and Paraguay (0.93 (0.88 – 1.0) Tg a$^{-1}$). Our estimates align with UNFCCC reports for Brazil, Bolivia, and Paraguay, but are significantly higher for other countries. Emissions in all countries are dominated by livestock (mainly enteric fermentation) except for oil/gas in Venezuela and landfills in Peru. Methane intensities from the oil/gas industry are high in Venezuela (33%), Colombia (6.5%) and Argentina (5.9%). Country-average emission factors for enteric fermentation from cattle in UNFCCC



reports are in the range 46-60 kg head$^{-1}$ a$^{-1}$, close to the IPCC Tier 1 estimate which is mostly based on data from Brazil. Our

inversion yields cattle enteric fermentation emission factors consistent with the UNFCCC reports for Brazil and Bolivia but a factor of two higher for other countries. The discrepancy for Argentina can be corrected by using IPCC Tier 2 emission estimates accounting for high milk production.

## 1 Introduction

Methane ($CH_4$) is a potent greenhouse gas with a relatively short atmospheric lifetime of $9.1 \pm 0.9$ years (Szopa et al., 2021).

Methane atmospheric concentrations have nearly tripled since pre-industrial times, resulting in an emission-based radiative forcing of $1.21$ W m$^{-2}$ compared to $2.16$ W m$^{-2}$ for $CO_2$ (Naik et al., 2021). Here we use satellite observations to quantify and attribute methane emissions from South American countries, which have been estimated to contribute 14% of global anthropogenic methane emissions (Worden et al., 2022) and are thought to be a major contributor to the methane rise over the past decade (Y. Zhang et al., 2021).

The 194 Parties to the Paris Agreement, including all 12 South American countries, must regularly submit Nationally Determined Contributions (NDCs) outlining their plans to reduce greenhouse gas emissions. These NDCs are based on national emission inventories constructed using bottom-up methods that combine activity data for individual sectors with emission factors, sometimes supplemented by direct measurements of individual sources. The inventories tend to have large uncertainties because emission factors (and sometimes the activity data) are poorly quantified (Saunois et al., 2020), and

even direct emission measurements may not capture source variability. Top-down information from atmospheric observations of methane concentrations can reduce these uncertainties through inverse analyses with an atmospheric transport model and using the bottom-up inventories as prior estimates in the inversion.

Anthropogenic emissions of methane come from many sectors, including oil/gas, coal, livestock, rice cultivation, landfills, and wastewater treatment. Natural emissions are from wetlands, fires, termites, and geological seeps. South American

anthropogenic emissions are heavily dominated by livestock. Of particular importance is Brazil, which is estimated to be the 3$^{rd}$ highest anthropogenic methane-emitting country globally (Worden et al., 2022) and has been identified as a major contributor to the recent global rise in methane through livestock and wetland emissions (Y. Zhang et al., 2021, Qu et al., 2024). Venezuela, Colombia, and Argentina are also high emitters (Worden et al., 2022). Wetlands are a major natural methane source in South America but again with large uncertainty (B. Zhang et al., 2017).

Satellite observations in the shortwave infrared (SWIR) are particularly attractive for top-down emission estimates due to their global coverage and sensitivity down to the surface. Inversions of data from the Greenhouse Gases Observing Satellite (GOSAT, 2009-present) (Parker et al., 2020a) have been used to infer the distribution of methane emissions globally (Maasakkers et al., 2019; Janardanan et al., 2020; Qu et al., 2021) and regionally for South America (Tunnicliffe et al., 2020, Wilson et al., 2021). Regional inversions have identified upward corrections in emissions inventories over Venezuela and the

Eastern Amazon, and downward corrections over the Western Amazon. However, GOSAT observations are sparse,



separated by about 250 km, which limits the spatial resolution that can be achieved, increasing uncertainties in attributing emissions to countries and sectors. The TROPOspheric Monitoring Instrument (TROPOMI) (2018-present) provides global continuous daily mapping of atmospheric methane at 7 km × 5.5 km nadir resolution (Lorente et al., 2023). This coverage in combination with high resolution provides TROPOMI with a unique capability for quantifying national emissions and attributing emissions to sectors. This has recently been demonstrated for the United States (Nesser et al., 2024), the Middle East and North Africa (Chen et al., 2023), China (Chen et al., 2022, Liang et al., 2023), and Venezuela (Nathan et al., 2023). Here we use TROPOMI observations in an inverse analysis of 2021 methane emissions over South America at up to 25 km resolution, using as prior estimates the national anthropogenic inventories reported to the United Nations Framework Convention on Climate Change (UNFCCC) under the Paris Agreement. We use two alternative bottom-up wetland emission inventories as prior estimates. We use a new TROPOMI satellite product that corrects surface, aerosol, and cloud artifacts with a machine learning algorithm trained by GOSAT data (Balasus et al., 2023). We also use GOSAT data, which though sparse provides unique information over wetlands. We quantify emissions by country and by sector and give recommendations for improving the bottom-up national inventories.

## 2 Data and Methods

We use methane observations from GOSAT and TROPOMI (Sect. 2.1) with the GEOS-Chem chemical transport model to optimize a state vector of mean methane emissions for 2021 over a rectilinear inversion domain covering South America (-57.5° to 13.25° latitude, -82.8125 ° to -33.75° longitude) at up to 0.25°×0.3125° resolution (~25 km × 25 km). We use countries' UNFCCC reports as prior estimates of anthropogenic emissions in our inversion such that our results directly evaluate those inventories (Sect. 2.2). We obtain posterior estimates of the state vector and the associated error covariance matrix though analytical solution for the minimum of the Bayesian cost function with lognormal prior errors (Sect. 2.3). We attribute inversion results to different methane emission sectors with the methodology described in Sect. 2.4. We conduct an ensemble of sensitivity inversions varying inversion parameters, including the choice of wetland prior estimate to characterize related errors in the posterior estimate (Sect. 2.5).

## 2.1 TROPOMI and GOSAT satellite observations

GOSAT, launched in 2009, has a 13:00 local overpass time and 10-km diameter pixels separated by about 250 km along-track and cross-track (Parker et al., 2020a). Dry column methane mixing ratios ($XCH_4$) are retrieved in the 1.65 μm absorption band with a $CO_2$ proxy method (Parker et al., 2011). The observations include a glint mode over the oceans. The $CO_2$ proxy method corrects for most surface and aerosol artifacts, yielding a global retrieval success rate of 23.5% limited by cloud cover (Parker et al., 2020a). We use the GOSAT v9.0 proxy retrieval from Parker and Boesch (2020) which is available at https://dx.doi.org/10.5285/18ef8247f52a4cb6a14013f8235cc1eb. We remove GOSAT observations in mountainous areas defined by a standard deviation of surface altitude greater than 25 m within a pixel as reported in the



GOSAT product. We also subtract 9.2 ppb from all GOSAT observations following Balasus et al. (2023) to remove the global mean bias versus TCCON. This yields $m_{GOSAT} = 29,233$ observations for 2021 used in our inversion.

TROPOMI is on board the polar sun-synchronous Sentinel 5 Precursor satellite launched in 2017 with a 13:30 local overpass time, providing full global daily coverage with a spatial resolution of 7 km × 5.5 km in the nadir (Veefkind et al., 2012). It retrieves $XCH_4$ with a full-physics algorithm in the 2.3 μm absorption band in combination with the NIR (757 – 774 nm) band. Again, the observations include a glint mode over the oceans. The global success rate is 3% over land limited by dark or heterogeneous surfaces and cloud cover (Hasekamp et al., 2023). The TROPOMI $XCH_4$ data can be affected by retrieval artifacts correlated with SWIR surface albedo (Lorente et al., 2023). Here we use the TROPOMI product from Balasus et al.

(2023), which uses a machine learning model to correct the TROPOMI v02.04.00 operational product of Lorente et al. (2023) by reference to GOSAT. The product is available at https://registry.opendata.aws/blended-tropomi-gosat-methane. There are 7,264,168 successful TROPOMI retrievals over the inversion domain during 2021. We average them over GEOS-Chem 0.25º×0.3125º grid cells to produce 885,957 super-observations (Chen et al., 2023). We filter out TROPOMI observations in grid cells that have fewer than 30 individual TROPOMI retrievals in 2021. This yields $m_{TROPOMI} = 853,599$

super-observations for 2021 used in the inversion.

Figure 1 shows the resulting data for TROPOMI and GOSAT in 2021 as the mean $XCH_4$ enhancements after subtracting the time- and latitude-dependent background over the oceans used as boundary conditions in the inversion (Section 2.3). Subtracting the background is needed for visualization because of its 100 ppb latitudinal difference between the northern and southern tips of South America. We see significant $XCH_4$ enhancements over wetlands, livestock regions, and urban areas.

There are few observations over the mountainous Andes, affecting much of Chile and Peru, so that the inversion for those countries relies significantly on glint observations offshore and on observations of transported methane. We also see that because of its use of the $CO_2$ proxy method, GOSAT is of particular value over the Amazon, where TROPOMI data are almost absent because of clouds and dark surfaces. Satellite observations are distributed throughout the year but are densest during the southern hemisphere dry season (June-September) (Figure 1).



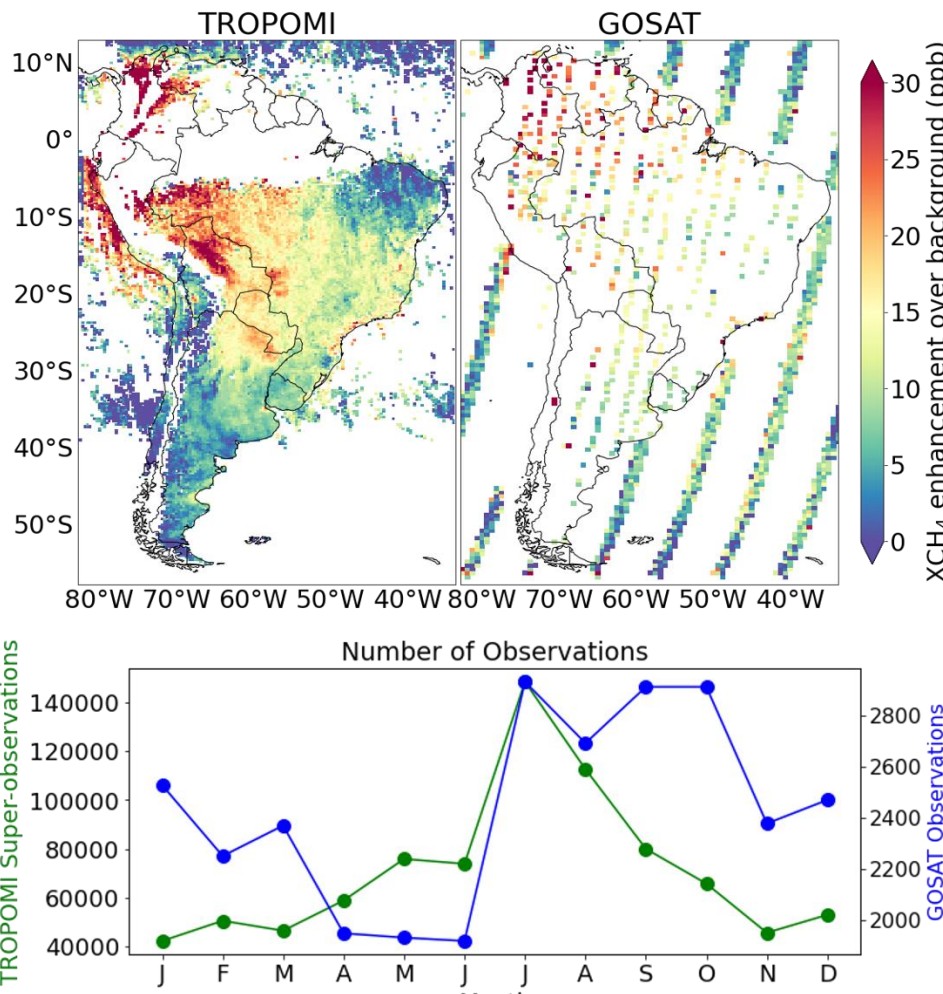

**Figure 1: Atmospheric methane enhancements observed by TROPOMI and GOSAT over South America relative to the latitudinal background. The Figure shows the mean 2021 dry-column methane mixing ratios (XCH$_4$) after subtraction of time- and latitude-dependent background values over the oceans used as boundary conditions in the inversion. TROPOMI observations are on the native grid of the inversion (0.25°×0.3125°) and GOSAT points are shown on a 0.5°×0.625° grid for visibility. GOSAT samples repeatedly at the same locations, partly accounting for the apparent sparsity. Also shown in the lower panel is the distribution of GOSAT observations and TROPOMI super-observations over the course of the year.**

## 2.2 Prior emissions

Fig. 2 shows the spatial distribution of prior emissions by sector on the 0.25°×0.3125° grid. Table 1 lists continental totals. Oil, gas, and coal emissions are from the Global Fuel Exploitation Inventory (GFEIv2) of Scarpelli et al. (2022), which uses detailed infrastructure data to spatially allocate emissions from countries' UNFCCC reports. National livestock, waste, and rice emissions are taken from each country's latest UNFCCC report (Table 2) and spatially distributed following the



Emissions Database for Global Atmospheric Research (EDGARv7) inventory for 2021 (Crippa et al., 2022). Other minor anthropogenic emissions are taken from EDGARv7. Anthropogenic emissions are assumed aseasonal except for rice, for which we use month-to-month variability from EDGARv6 (Crippa et al., 2021).

UNFCCC national totals listed in Table 2 and taken as prior estimates for the inversion are from the UNFCCC GHG Data Interface (https://di.unfccc.int/detailed_data_by_party, last accessed Jan 20, 2023). Some countries are not available in the UNFCCC GHG Data Interface, and we inspect reports submitted by each country including National Communications (https://unfccc.int/non-annex-I-NCs, last accessed Jan 20, 2023) and Biennial Update Reports (https://unfccc.int/BURs, last accessed Jan 20, 2023), to obtain the most recent emissions estimates. These reports contain emissions estimates for the

following years: Argentina (report published in 2022, emissions provided for 2018), Brazil (2020, 2016), Bolivia (2020, 2008), Chile (2022, 2020), Colombia (2022, 2018), Ecuador (2023, 2018), Guyana (2012, 2004), Paraguay (2022, 2017), Peru (2019, 2014), Suriname (2022, 2017), Uruguay (2022, 2019), and Venezuela (2018, 2010). French Guiana is not independently reported and we use GFEIv2 for fuel and EDGARv7 for all other sectors.

Two alternative monthly wetland emission inventories for 2021 with 0.5°×0.5° spatial resolution are used as prior estimates:

WetCHARTs and LPJ-wsl. WetCHARTs is an ensemble of parameterized inventories applying different inundation data, temperature dependence, and other factors (Bloom et al., 2017). We use the mean emissions from the nine high-performance members of the WetCHARTs v1.3.1 ensemble found by Ma et al. (2021) to best fit the results from a global GOSAT inversion and refer to it as WetCHARTs in what follows. LPJ-wsl is based on the Dynamic Global Vegetation model (Z. Zhang et al., 2016) driven here with NASA MERRA-2 meteorological data (Z. Zhang et al., 2018) and is henceforth referred

to as LPJ-MERRA2. East et al. (2024) found that LPJ-MERRA2 could reproduce seasonal variations of atmospheric methane concentrations better than other wetland inventories, including WetCHARTS.

Other natural sources in our prior estimates include daily open-fire emissions for 2021 from the Global Fire Emissions Database version 4s (GFED4s) (van der Werf et al., 2017), termite emissions from Fung et al. (1991), and geological seepage emissions from Etiope et al. (2019) with global scaling to 2 Tg a$^{-1}$ (Hmiel et al., 2020).

Figure 2 shows that South American emissions in the prior estimate are dominated by wetlands (62% of continental emissions averaged across LPJ-MERRA2 and WetCHARTs), mainly over the Amazon region but also extending into Paraguay (Pantanal) and Argentina (Paraná). Livestock (22%), mainly enteric fermentation from cattle, is the largest anthropogenic source for almost all countries and is spatially distinct from wetlands. Landfills and wastewater treatment, collectively referred to as waste (5.9%), follow population density and are large in all countries. Fossil fuel emissions are

mostly from oil/gas (2.5%) and are concentrated in Venezuela and Argentina. Coal emissions are small (0.4%) and concentrated in Colombia. Rice emissions are also small (0.7%) and concentrated in southernmost Brazil. Open fires are a large seasonal source (2.5%) concentrated along the southern edge of the Amazon in Brazil and northern Bolivia.



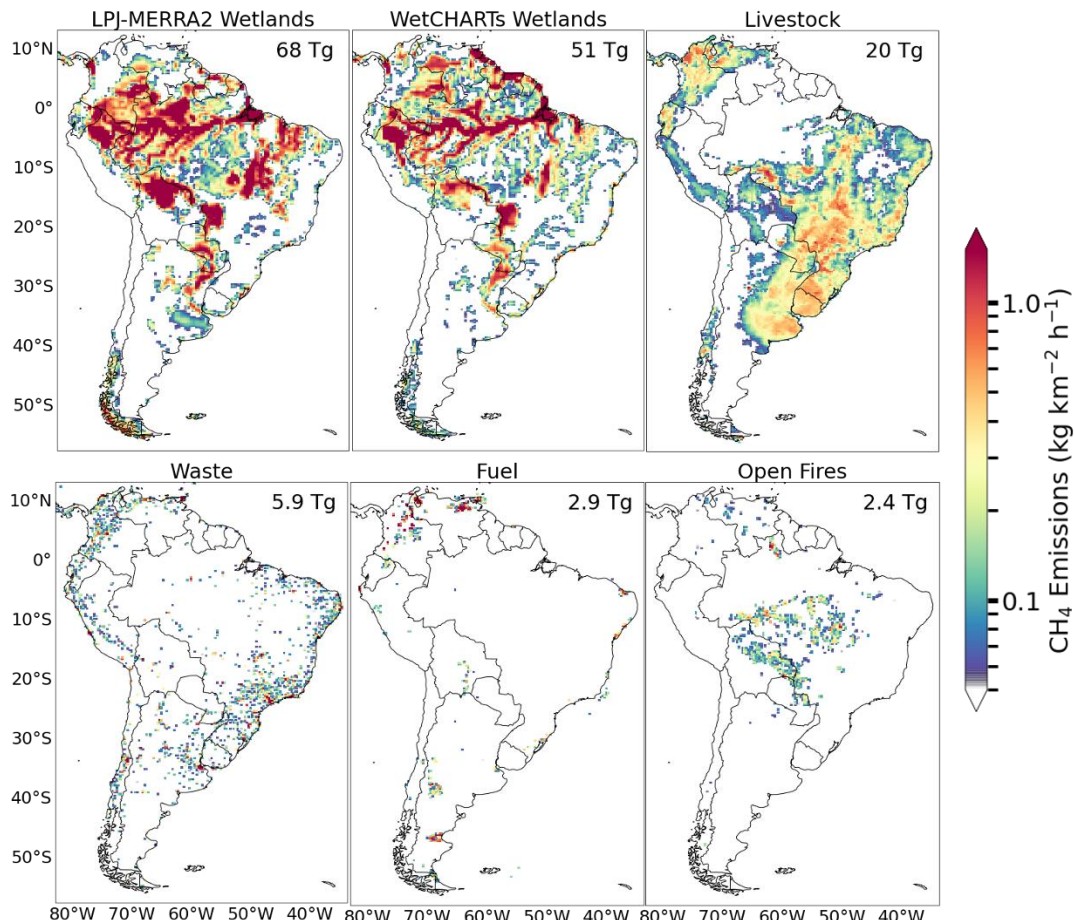

**Figure 2: Bottom-up methane emission inventories used as prior estimates for the inversion. Panels show annual mean methane emissions for major sectors with continental totals inset. Wetland emissions for 2021 (inversion year) are shown for both WetCHARTs (mean of the nine members of the high-performing ensemble) and LPJ-MERRA2. Coal, oil, and gas emissions are from the GFEIv2 gridded version of the national inventories from individual countries reported to the UNFCCC. Other anthropogenic emissions are from countries' most recent UNFCCC reports with spatial allocation from EDGAR v7. See Table 2 for national totals as reported by individual countries to the UNFCCC for years ranging from 2004 to 2020.**



**Table 1: Methane emissions in South America (Tg a⁻¹)**

|  | Prior[a] | Posterior[b] |
|---|---|---|
| **Total** | 96 | 121 (109-137) |
| **Anthropogenic (UNFCCC)** | 31 | 48 (41-56) |
| **Livestock** | 21 | 31 (27-37) |
| **Waste[c]** | 5.7 | 7.8 (6.5-9.5) |
| **Rice** | 0.68 | 0.86 (0.74-1.4) |
| **Oil and Gas** | 2.4 | 6.2 (5.2-7.9) |
| **Coal** | 0.40 | 0.59 (0.42-1.4) |
| **Other[d]** | 1.0 | 1.3 (1.1-1.5) |
| **Natural** | 56-74 | 74 (68-83) |
| **Open Fires** | 2.4 | 2.6 (2.4-3.0) |
| Wetlands | 52-68[e] | 67 (62-75) |
| **Seeps** | 0.09 | 0.22 (0.17-0.30) |
| **Termites** | 2.6 | 3.8 (3.2-5.1) |

[a] Prior emission estimates used in the inversion. Anthropogenic emissions are from national reports to the UNFCCC for years ranging from 2004 to 2020 (see Table 2 for individual countries). Wetland and open fire emissions are for 2021 (inversion year).

[b] Median best estimates from the inversion of TROPOMI and GOSAT data for 2021, and ranges from the inversion ensemble.

[c] Including landfills and wastewater treatment

[d] Including industry, stationary combustion, mobile combustion, aircraft, composting, and field burning of agricultural residues.

[e] Prior estimates for the inversion are taken either from the mean of the high-performing subset of the WetCHARTs ensemble (52 Tg a⁻¹) or from LPJ-MERRA2 (68 Tg a⁻¹).

## 2.3 Analytical Inversion

We use the Integrated Methane Inversion workflow (IMI 1.1) (Varon et al., 2022) with modifications. The forward model for the inversion is the nested version of the GEOS-Chem 14.1.1 chemical transport model (https://doi.org/10.5281/zenodo.4618180), which relates methane emissions to atmospheric concentrations through atmospheric transport (Maasakkers et al., 2019). GEOS-Chem is driven by meteorological fields from NASA GEOS-FP analyses at $0.25° \times 0.3125°$ resolution. We use this native resolution in GEOS-Chem over South America and adjacent oceans (domain in Fig. 1) with dynamic boundary conditions outside the inversion domain updated every 3 h from a global archive of smoothed TROPOMI observations (Varon et al., 2022). That same archive is used as initial conditions, so that the simulation is initially unbiased relative to TROPOMI observations. The GEOS-Chem methane simulation includes chemical loss from oxidation by tropospheric OH with a corresponding methane lifetime of 10.8 years, consistent with the lifetime of $11.2 \pm 1.3$ years inferred from the methyl chloroform proxy (Prather et al., 2012). It also includes minor losses from



oxidation by tropospheric Cl, oxidation in the stratosphere, and uptake by soils (Murguia-Flores et al., 2018). We do not
optimize these sinks here.

We select the state vector $x$ for the inversion with the Gaussian mixture model (GMM) of Turner and Jacob (2015) modified
to include satellite observation density as a similarity criterion. The GMM selects emission patterns that the TROPOMI
observations can effectively constrain, aiming to preserve native (0.25°× 0.3125°) resolution for strong sources with high
observation density while smoothing the solution in regions with low observation density or weak prior emissions. Similarity
vectors defining proximity and commonality in sectoral emissions (as defined by the prior estimate) as well as the density of
TROPOMI observations are used to construct Gaussian state vector elements. We use 600 Gaussian functions as state vector
elements to balance aggregation and smoothing errors (Wecht et al., 2014). We also optimize boundary conditions for each
quadrant (north, south, west, east) and for each season, for a total of $n = 616$ state vector elements.

We perform the inversion with lognormal error probability density functions (pdfs) for prior emissions. This prevents
unphysical negative emissions and better captures the heavy tail of the emission distribution than a normal error assumption.
Specifically, we optimize $\ln(x)$ instead of $x$, such that the prior errors on $\ln(x)$ (henceforth denoted as $x'$) follow a normal
distribution. We optimize the boundary condition elements of the state vector assuming normal error distributions.

The inversion finds the optimal estimate $\hat{x}'$ of $x'$ assuming normal error distributions (lognormal in emission space) by
minimizing the Bayesian cost function $J(x')$ (Brasseur and Jacob, 2017):

$$J(x') = (x' - x'_a)^T S_a'^{-1}(x' - x'_a) + \gamma (y - K'x')^T S_o^{-1}(y - K'x'),\tag{2}$$

where $x' = \ln(x)$ and $x'_a = \ln(x_a)$, $x_a$ ($n \times 1$) is the prior emission estimate ($n = 616$), and $y$ ($m \times 1$) is the ensemble of
TROPOMI super-observations and GOSAT observations. $S_a'$ ($n \times n$) is the prior error covariance matrix and $S_o$ ($m \times m$) is
the observational error covariance matrix. We assume $S_a'$ and $S_o$ to be diagonal in absence of better objective information.
$K'x' = Kx$ is the GEOS-Chem forward model simulation of XCH$_4$ which is constructed from the GEOS-Chem vertical
profiles of methane dry mixing ratios by applying TROPOMI or GOSAT averaging kernel vectors and prior vertical profiles.
$K = \frac{\partial y}{\partial x}$ ($m \times n$) is the Jacobian matrix that describes the linear sensitivity of $y$ to $x$, and is constructed column by column by
perturbing individual elements of $x$ in GEOS-Chem. $K' = \frac{\partial y}{\partial x'}$ ($m \times n$) describes the sensitivity of $y$ to $x'$, which is
nonlinear but derived immediately from $K$ with matrix elements $k'_{i,j} = \frac{\partial y_i}{\partial \ln(x_j)} = x_j \frac{\partial y_i}{\partial x_j} = x_j k_{i,j}$ where $i$ and $j$ are indices of
the observations and the state vector elements respectively. The regularization factor $\gamma$ is used to prevent overfitting to
observations caused by missing covariant structure (off-diagonal terms) in $S_o$ (Chevallier, 2007). Following the method of
Lu et al. (2021), we determine an optimal $\gamma$ value such that $(\hat{x}' - x'_a)^T S_a'^{-1}(\hat{x}' - x'_a) \approx n \pm \sqrt{2n}$, the expected value (±1
standard deviation) of the Chi-square distribution with $n$ degrees of freedom. This yields $\gamma = 0.05$ here.

We solve the nonlinear optimization problem iteratively using the Levenberg-Marquardt method (Rodgers, 2000):

$$x'_{N+1} = x'_N + \left(\gamma K_N'^T S_o^{-1} K_N' + (1+\kappa)S_a'^{-1}\right)^{-1}\left(\gamma K_N'^T S_o^{-1}(y - Kx_N) - S_a'^{-1}(x'_N - x'_a)\right),\tag{2}$$



where the coefficient κ is fixed at 10 following Chen et al. (2022), who found that using κ = 10 converges faster with no difference in results compared to other methods. $N$ is the iteration number with $\boldsymbol{x}'_0 = \boldsymbol{x}'_a$, and $\mathbf{K}'_N$ is evaluated for $\boldsymbol{x}' = \boldsymbol{x}'_N$. We iterate on Eq. (2) until the differences of all state vector elements between two consecutive iterations ($\boldsymbol{x}'_N$ and $\boldsymbol{x}'_{N+1}$) are smaller than 0.5% and then take $\hat{\mathbf{x}}' = \mathbf{x}'_{N+1}$ as the optimal posterior estimate. The posterior error covariance matrix $\hat{\mathbf{S}}'$ on the optimal posterior estimate is given by (Rodgers, 2000):

$$\hat{\mathbf{S}}' = (\gamma \mathbf{K}'^T \mathbf{S}_o^{-1} \mathbf{K}' + \mathbf{S}_a'^{-1})^{-1} \ , \tag{3}$$

where $\mathbf{K}' = \mathbf{K}'_{N+1}$ is evaluated for the posterior estimate. The averaging kernel matrix $\mathbf{A}$ defining the sensitivity of the solution to the true value is given by:

$$\mathbf{A} = \frac{\partial \hat{x}'}{\partial x'} = \mathbf{I}_n - \hat{\mathbf{S}}' \mathbf{S}_a'^{-1} \ , \tag{4}$$

where $\mathbf{I}_n$ is the $n \times n$ identity matrix. The trace of $\mathbf{A}$, which is called the degrees of freedom for signal (DOFS), indicates the number of independent pieces of information on $\boldsymbol{x}'$ obtained from the observations. We will refer to the averaging kernel sensitivity for individual state vector elements as the corresponding diagonal element of the averaging kernel matrix.

An implication of using lognormal error statistics for emissions is that the prior estimate $\boldsymbol{x}_a$ is the median (not the mean) of a lognormal error pdf, and the inversion correspondingly optimizes the median of the lognormal emission pdf. But the UNFCCC national reports should be viewed as best prior estimates of the means of the emission pdfs since they are to be added across countries for the Global Stocktake. The median and the mean of a lognormal pdf are related by:

$$x_{\text{median}} = x_{\text{mean}} \exp\left[-\frac{s'}{2}\right], \tag{5}$$

where $s' = (\ln \sigma_g)^2$ is the error variance in normal space and $\sigma_g$ is the geometric error standard deviation. Here we assume that the prior emissions are lognormally distributed with a geometric standard deviation of 2 ($\sigma_g = 2$), therefore $x_{\text{median}} = 0.79 x_{\text{mean}}$. We apply these corrections to the prior emission estimates from Section 2.2 for use in the inversion as $\boldsymbol{x}_a$, with the prior error covariance matrix $\mathbf{S}_a'$ taken as a diagonal matrix of the error variances $s_a = (\ln 2)^2$.

The same operation in reverse is needed for interpreting the posterior emission estimates, which the inversion returns as the medians of the posterior lognormal error pdf with posterior error covariance matrix $\hat{\mathbf{S}}'$. From the posterior error variances $\hat{s}'_j$ given by the diagonal elements of $\hat{\mathbf{S}}'$ for the individual state vector elements $j$, we apply for each element the conversion $\hat{x}_{j,\text{mean}}, = \hat{x}_{j,\text{median}} \exp\left[\hat{s}'_j/2\right]$. The mean posterior estimates are therefore related to the mean prior estimates by:

$$\hat{x}_{j,\text{mean}} = \left(\frac{\hat{x}_j}{x_{j,a}}\right)_{\text{inversion}} \exp\left[\frac{\hat{s}'_j - s'_{j,a}}{2}\right] x_{j,a,\text{mean}} \ , \tag{6}$$

where $(\hat{x}_j / \hat{x}_{j,a})_{\text{inversion}}$ is the ratio of medians returned by the inversion. All results presented here are for the mean posterior estimates, which allows for the summing of inversion results geographically to obtain regional or national totals for comparison to the mean prior estimates. We set the prior error standard deviation on the boundary conditions to be 10 ppb, which is typical of the root mean square error (RMSE) of GEOS-Chem simulations using posterior emission estimates (Chen et al., 2022).





We use the residual error method (Heald et al., 2004) to estimate observational error variances including contributions from the TROPOMI and GOSAT instruments, the retrieval, and the forward model. This method takes the residual error between the observations and the forward model simulation with prior estimates (after removing the mean bias, to be corrected in the inversion) as measure of the observational error on the forward model grid. We do this separately for GOSAT and
265    TROPOMI. The resulting mean observational error standard deviation for GOSAT is 11.2 ppb. To account for the error reduction resulting from averaging $P$ individual TROPOMI retrievals into the super-observations $y$ on the GEOS-Chem $0.25° \times 0.3125°$ grid, we employ the residual error method for super-observations developed by Chen et al. (2023). This method derives the observational error variance of the super-observations ($\sigma^2_{super}$) by separating the contributions in the individual observations from the transport error variance $\sigma^2_{transport}$ (perfectly correlated for the individual observations within a GEOS-Chem grid cell) and the satellite single-retrieval error variance ($\sigma^2_{retrieval}$):

$$\sigma^2_{super} = \sigma^2_{retrieval}\left(\frac{1-r_{retrieval}}{P} + r_{retrieval}\right) + \sigma^2_{transport}, \tag{7}$$

where $r_{retrieval}$ is the error correlation coefficient for the individual observations in a $0.25° \times 0.3125°$ grid cell averaged into a super-observation. Chen et al. (2023) obtained $\sigma_{transport} = 4.5$ ppb, $\sigma_{retrieval} = 16.5$ ppb, and $r_{retrieval} = 0.55$ for TROPOMI observations over the Middle East and North Africa. Our own fit of residual errors to Eq. (7) for South America yields
275    $\sigma_{transport} = 4.3$ ppb, $\sigma_{retrieval} = 14.8$ ppb, and $r_{retrieval} = 0.21$. The average observational error standard deviation for the TROPOMI super-observations in the inversion domain is 7.9 ppb.

**2.4 Attributing posterior emissions to individual countries and sectors**

The posterior GMM state vector ($n \times 1$) can be mapped onto the $p$ native $0.25° \times 0.3125°$ grid cells of the inversion domain using the GMM-generated weighting of each Gaussian on that grid as represented by a matrix $\mathbf{W_{GMM}}$ ($p \times n$). The
280    contributions from each of $q$ emission sectors to the emissions in individual grid cells are taken from the prior inventories to produce a matrix $\mathbf{W_{sectors}}$ ($pq \times n$). We then apply a summation matrix $\mathbf{W_{agg}}$ ($r \times pq$) to aggregate emissions over $r$ countries or sectors of interest. The resulting matrix $\mathbf{W} = \mathbf{W_{agg}W_{sectors}}$ ($r \times n$) represents the linear transformation from the posterior GMM state vector ($n \times 1$) to a reduced state vector ($r \times 1$) of sectoral or country-level emissions. The reduced state vector ($\hat{\mathbf{x}}_{red}$), posterior error covariance ($\hat{\mathbf{S}}_{red}$), and averaging kernel matrix ($\mathbf{A}_{red}$) are computed as:

$$\hat{\mathbf{x}}_{red} = \mathbf{W}\hat{\mathbf{x}}, \tag{8}$$

$$\hat{\mathbf{S}}_{red} = \mathbf{W}\hat{\mathbf{S}}\mathbf{W^T}, \tag{9}$$

$$\mathbf{A}_{red} = \mathbf{W}\mathbf{A}\mathbf{W}^*, \tag{10}$$

where $\mathbf{W}^* = \mathbf{W^T}(\mathbf{WW^T})^{-1}$ is the Moore-Penrose pseudo-inverse of $\mathbf{W}$ (Calisesi et al., 2005). We either aggregate together or make note of sectors that have an error correlation greater than 0.75 as given by $\hat{\mathbf{S}}_{red}$. The averaging kernel sensitivities
290    for the aggregated emissions are the diagonal elements of $\mathbf{A}_{red}$ and represent the ability of the inversion to quantify the emissions independently from the prior estimate (1 = fully, 0 = not at all).



## 2.5 Inversion Ensemble

Our inversion described above makes assumptions on the values of inversion parameters including a geometric error standard deviation of the lognormal prior error distribution $\sigma_g = 2$, an error standard deviation $\sigma_b = 10$ ppb for boundary conditions, and a regularization factor $\gamma = 0.05$. The posterior error matrix of Eq. (3) represents the uncertainty in the analytical solution given this choice of inversion parameters, but it does not account for uncertainties in the parameters themselves, including the prior emission estimate. The choice of wetland emission inventory used as prior estimate for the inversion could particularly affect results. To address this, we generate a 54-member ensemble of sensitivity inversions varying the parameters following Chen et al. (2023). The inversion ensemble includes (1) $\sigma_g = 1.5$, 2, or 2.5, (2) $\sigma_b = 5$, 10, or 20 ppb, (3) WetCHARTS or LPJ-MERRA2 wetland prior estimate, and (4) $\gamma = 0.025$, 0.05, or 0.1. Because the uncertainty defined by the range of optimal estimates of this ensemble is larger than the posterior error from any single inversion, we report a conservative uncertainty in posterior estimates as the range of solutions given by the inversion ensemble. Unless stated otherwise, we report the best posterior estimate of emissions as the median of this inversion ensemble (for each state vector element, prior emissions are scaled by the median posterior/prior emissions ratio across the ensemble).

## 3 Results and Discussion

### 3.1 Continental-scale Results

Figure 3 shows the prior and posterior emission estimates over the continental scale for the median of the inversion ensemble, along with the median averaging kernel sensitivities. The median DOFS (sum of the averaging kernel sensitivities) are 144, out of a maximum of 616 defined by the state vector dimension. Low averaging kernel sensitivities over the Amazon reflect the sparsity of observations. The inversion effectively fits the emissions to the satellite data, as shown in Figure 4 where posterior emissions decrease the mean GEOS-Chem model bias relative to the observations over the inversion domain from 3.04 to -0.03 ppb. The root-mean-square error (RMSE) decreases from 9.65 to 8.53 ppb, with improvement limited by the observational error (7.9 ppb for TROPOMI and 11.2 ppb for GOSAT).



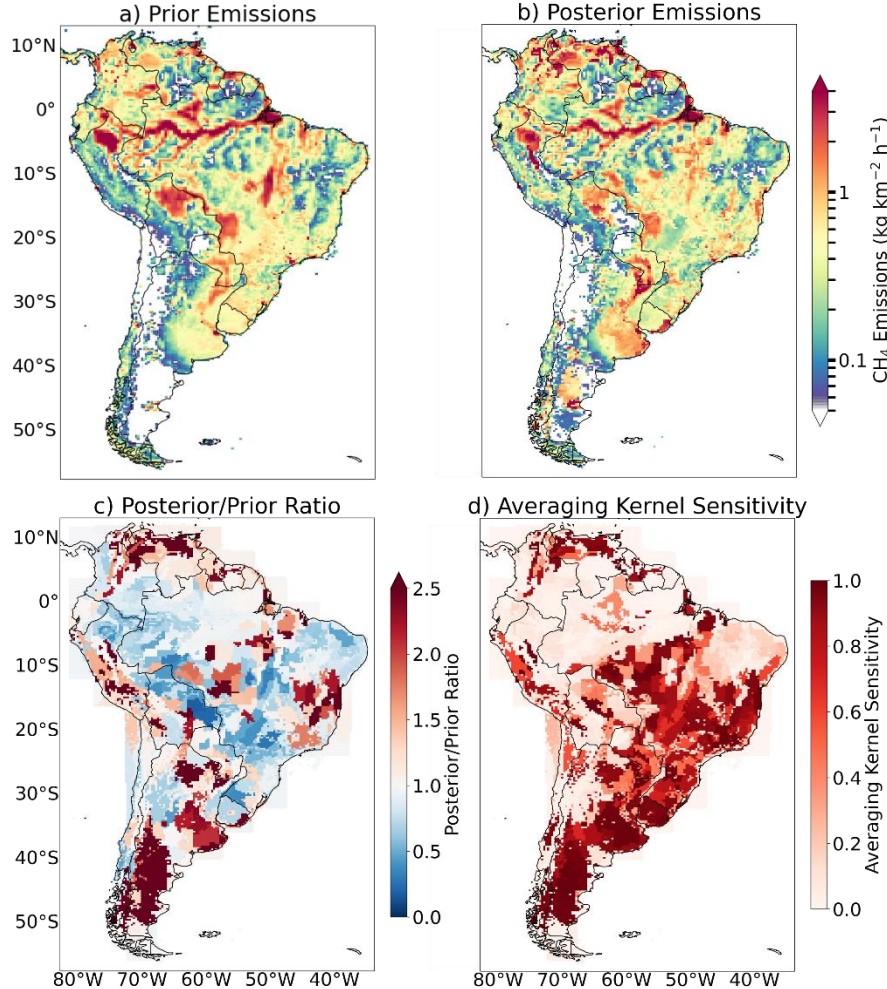

**Figure 3: Optimization of methane emissions in South America on the 0.25°×0.3125° grid. Posterior emissions are our best estimates from the inversion of TROPOMI+GOSAT observations. Prior estimates are from UNFCCC reports for anthropogenic emissions and either LPJ-MERRA2 or WetCHARTs for wetland emissions (Figure 2: the average is shown here). The averaging kernel sensitivities indicate the ability of the observations to quantify emissions independently from the prior estimates on the 0.25°×0.3125° grid (1 = fully; 0 = not at all) as given by the diagonal elements of the averaging kernel matrix.**

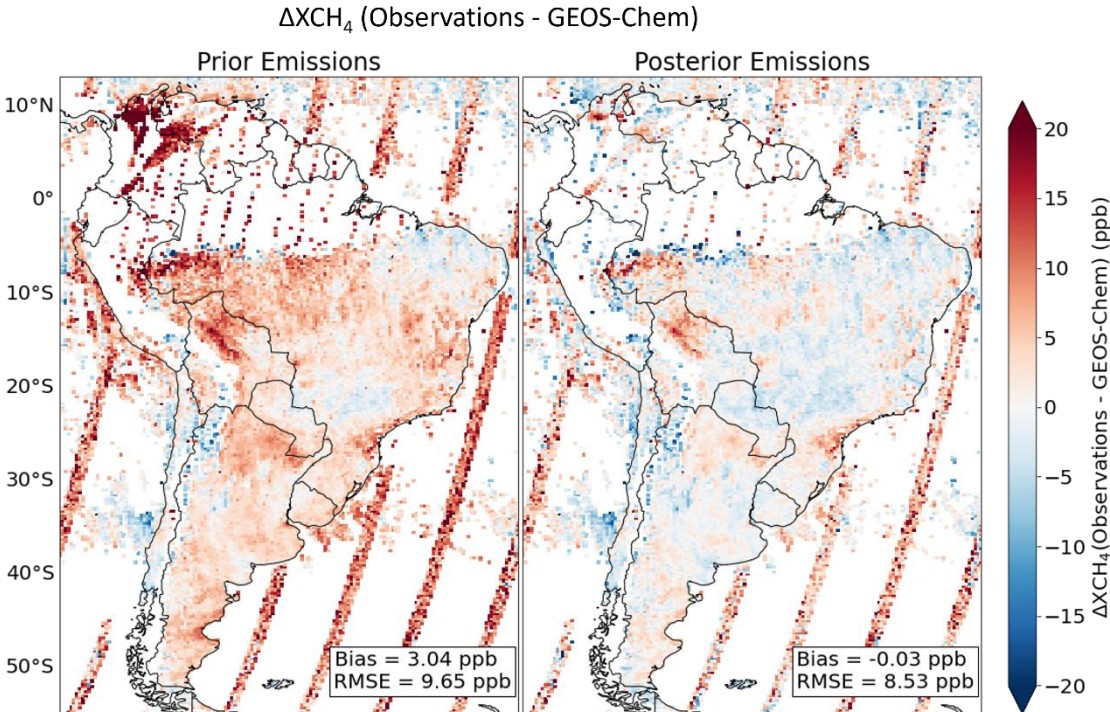

**Figure 4: Differences between methane dry column mixing ratios (XCH₄) observed by TROPOMI+GOSAT and simulated by GEOS-Chem with prior emissions (including wetland emissions averaged across WetCHARTs and LPJ-MERRA2) and posterior emissions (median of the inversion ensemble). Legends give the mean bias and root-mean-square errors (RMSEs) for the prior and posterior.**

Table 1 compares total prior and posterior emission estimates for South America. Posterior emissions are 121 (109-137) Tg a$^{-1}$, where the parentheses indicate the range from the inversion ensemble. This represents a significant increase from the prior estimate of 96 Tg a$^{-1}$. Most of that increase is from anthropogenic emissions, which increase from 31 Tg a$^{-1}$ in the UNFCCC reports used as prior estimates to 48 (41-56) Tg a$^{-1}$. All sectors show emissions increases, with the largest for oil/gas (158%). Further discussion of emissions by sector and country is presented below.

**3.2 Wetland emissions**

Figure 5 shows the difference between posterior and prior wetland emissions over South America from the WetCHARTs and LPJ-MERRA2 inversions with γ=0.05, σ$_g$=2, and σ$_b$=10 ppb. Inversion results are sensitive to the choice of prior estimate, even though the averaging kernel sensitivity is high, because of large differences in the prior spatial distributions (Figure 2). While the continental-scale correction to wetland emissions from the inversion is smaller for LPJ-MERRA2 (+0.5 Tg a$^{-1}$) than WetCHARTs (+15.9 Tg a$^{-1}$), the sum of the absolute value of spatial differences is larger for LPJ-MERRA2 (46 Tg a$^{-1}$) than WetCHARTs (36 Tg a$^{-1}$) (Figure 5). East et al. (2024) found that LPJ-MERRA2 better matched zonal mean



atmospheric observations than WetCHARTs, but we find here that the WetCHARTs spatial distribution over South America better matches our posterior emissions estimate.


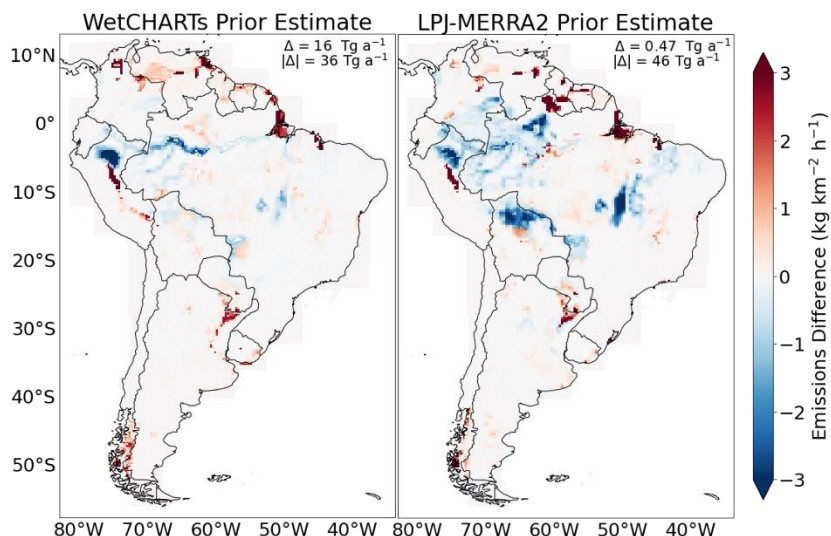

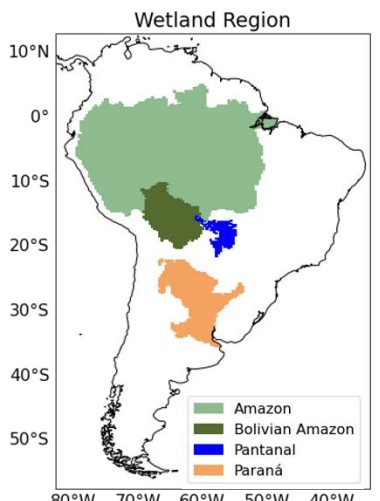

| Region | Wetland Inventory | Wetland emissions (Tg a$^{-1}$) | | Averaging Kernel Sensitivity |
|---|---|---|---|---|
| | | Prior Estimate | Posterior Estimate | |
| Amazon | WetCHARTs | 32 | 32 (29-37) | 0.74 |
| | LPJ-MERRA2 | 46 | 39 (34-44) | 0.73 |
| Bolivian Amazon | WetCHARTs | 1.9 | 1.7 (1.6 - 2.0) | 0.60 |
| | LPJ-MERRA2 | 7.0 | 3.8 (3.6-4.4) | 0.63 |
| Pantanal | WetCHARTs | 1.8 | 1.6 (1.5 - 1.8) | 0.87 |
| | LPJ-MERRA2 | 2.0 | 1.3 (1.2-1.5) | 0.76 |
| Paraná | WetCHARTs | 0.87 | 2.0 (1.8 - 2.1) | 0.92 |
| | LPJ-MERRA2 | 1.4 | 2.1 (2.0 - 2.2) | 0.89 |
| South America | WetCHARTs | 52 | 67 (62 - 73) | 0.77 |
| | LPJ-MERRA2 | 68 | 69 (65 - 75) | 0.74 |

**Figure 5: Correction to wetland emissions from inversion of TROPOMI and GOSAT data. The top panels show the differences between posterior and prior wetland emissions when either WetCHARTs or LPJ-MERRA2 wetland emissions are used as prior estimates. with γ=0.05, σ$_g$=2, and σ$_b$=10. The bottom panels show the prior and posterior wetland emissions for different regions.**

**Ranges from the inversion ensembles are in parentheses. Boundaries of each region are defined using a combination of hydrological basin data from FAO's AQUASTAT (AQUASTAT database, available at https://data.apps.fao.org/aquastat/?lang=en, last accessed February 2024), and terrestrial ecoregions from the World Wildlife Fund (Olson et al., 2001).**



Further examination of wetland emissions is shown in Figure 5 for four major regions: the Amazon Basin, the Bolivian Amazon, the Pantanal, and the Paraná. These regions constitute 68% and 83% of South American wetland emissions according to WetCHARTs and LPJ-MERRA2, respectively. We find emissions from the Amazon Basin of 32 (29-44) Tg a$^{-1}$, aligning with the WetCHARTs estimate and within the range of uncertainty of other estimates (31-56.5 Tg a$^{-1}$) (Wilson et al., 2021; Wilson et al., 2016; Ringeval et al., 2014; Pangala et al., 2017; Basso et al., 2021). The Bolivian Amazon is a

region of interest because of recent aircraft measurements showing methane emissions of 3.6 Tg a$^{-1}$ (France et al., 2022). Our best posterior estimate is 2.8 (1.6-4.4) Tg a$^{-1}$, again more consistent with WetCHARTs (1.9 Tg a$^{-1}$) than LPJ-MERRA2 (7 Tg a$^{-1}$).

The Pantanal, located below the Amazon basin in Brazil, Bolivia, and Paraguay, is the largest seasonally flooded tropical grassland in the world. We estimate emissions from the Pantanal to be 1.5 (1.2-1.8) Tg a$^{-1}$ with downward correction from

both LPJ-MERRA2 and WetCHARTs (1.8 and 2.0 Tg a$^{-1}$) and lower than the range of uncertainty of previous estimates (1.9 – 3.3 Tg a$^{-1}$) (Bastviken et al., 2010; Marani and Alvalá, 2007; Gloor et al., 2021).

The Paraná River wetland region extends from northern Argentina to the Paraná River Delta, which feeds into the Atlantic Ocean. We estimate emissions from this region to be 2.0 (1.8-2.2) Tg a$^{-1}$, a narrow range reflecting the high averaging kernel sensitivity. This is larger than WetCHARTs (0.87 Tg a$^{-1}$) and LPJ-MERRA2 (1.4 Tg a$^{-1}$). Parker et al. (2020b) found that

WetCHARTs underestimated Paraná emissions in comparison to GOSAT due to wetland extent underestimation.

**3.3 Anthropogenic emissions from individual countries and sectors**

Figure 6 shows emissions by sector from the top seven anthropogenic emitting countries that make up 90% of posterior anthropogenic emissions over South America. Table 2 shows emissions for all countries. Posterior error correlations between countries are all less than 0.25, indicating the inversion's ability to effectively separate emissions between countries, but

averaging kernel sensitivities are low (< 0.3) for Ecuador, French Guiana, and Suriname, because of a low density of observations and low prior emissions. We aggregate emissions from oil and gas as well as wastewater and landfills since posterior errors for these sectors are highly correlated. Posterior error correlations between other major sectors are generally low (< 0.25). Livestock has higher error correlations with rice (0.42) and biomass burning (0.44), but these are small sources.

We find that UNFCCC-reported emissions for Brazil, Bolivia, and Paraguay are within the range of our inversion

ensemble while UNFCCC-reported emissions for Argentina, Venezuela, Colombia, and Peru are too low. Livestock emissions are underestimated in all four of these countries. Argentina and Venezuela also underestimate their oil/gas emissions. Peru has a large contribution from waste emission that is underestimated in its UNFCCC report. Nathan et al. (2023) conducted a regional TROPOMI inversion over Venezuela and found total anthropogenic emissions in 2019 to be 3.6 (2.0-5.3) Tg a$^{-1}$. This is much lower than our estimate of 7.0 (5.5 – 9.9) mainly from differences in emissions from livestock

(1.2 (0.9-1.6) Tg a$^{-1}$ vs our posterior 2.8 (2.0-4.5) Tg a$^{-1}$) and oil/gas (1.8 (0.9-2.7) Tg a$^{-1}$ vs 3.4 (3.1-5.5) Tg a$^{-1}$). Their lower estimate may be due to low averaging kernel sensitivities from using TROPOMI alone that restrict significant departure from the prior estimate.





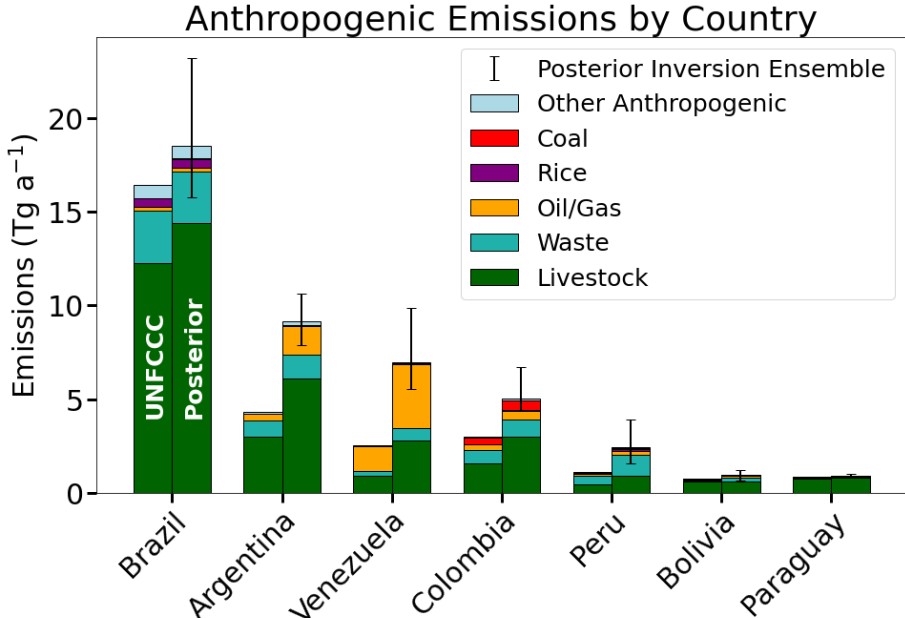

**Figure 6: National anthropogenic methane emissions from the top seven emitting countries in South America. Posterior estimates**
**from inversion of 2021 TROPOMI and GOSAT observations are compared to the most recent UNFCCC reports taken as prior estimates for the inversion (Table 2). Waste includes emissions from landfills and wastewater treatment, which cannot be separated by the inversion. Vertical lines show the range of posterior estimates from our inversion ensemble.**



**Table 2: National anthropogenic emissions (Tg a[-1]) by country and sector[a]**

| Country | Total anthropogenic | Livestock | Waste | Rice | Oil/Gas | Coal | Other[b] | Averaging kernel sensitivity[c] |
|---|---|---|---|---|---|---|---|---|
| **Argentina** | | | | | | | | |
| UNFCCC | 4.3 | 3.0 | 0.82 | 0.02 | 0.36 | <0.01 | 0.12 | |
| Posterior | 9.2 (7.9-11) | 6.1 (5.1-7.1) | 1.3 (1.1-1.6) | 0.02 (0.02-0.03) | 1.5 (1.4-1.6) | <0.01 | 0.21 (0.18-0.25) | 0.94 |
| **Bolivia** | | | | | | | | |
| UNFCCC | 0.75 | 0.58 | 0.09 | 0.02 | 0.05 | <0.01 | 0.01 | |
| Posterior | 0.96 (0.66-1.2) | 0.61 (0.47-0.77) | 0.18 (0.11-0.20) | 0.02 (0.01-0.02) | 0.13 (0.07-0.19) | 0.18 (0.11-0.19) | 0.02 (0.01-0.02) | 0.6 |
| **Brazil** | | | | | | | | |
| UNFCCC | 16 | 12 | 2.8 | 0.44 | 0.18 | 0.04 | 0.68 | |
| Posterior | 19 (16-23) | 14 (12-18) | 2.8 (2.4-3.3) | 0.49 (0.42-0.61) | 0.16 (0.13-0.17) | 0.04 (0.03-0.05) | 0.68 (0.58-0.84) | 0.75 |
| **Chile** | | | | | | | | |
| UNFCCC | 0.67 | 0.24 | 0.34 | <0.01 | 0.04 | <0.01 | 0.04 | |
| Posterior | 0.88 (0.69-0.96) | 0.36 (0.26-0.41) | 0.41 (0.34-0.44) | 0.01 (<0.01-0.01) | 0.05 (0.04-0.15) | <0.01 | 0.05 (0.04-0.05) | 0.61 |
| **Colombia** | | | | | | | | |
| UNFCCC | 3.0 | 1.6 | 0.69 | 0.03 | 0.28 | 0.35 | 0.05 | |
| Posterior | 5.0 (4.4-6.7) | 3.0 (2.5-4.2) | 0.91 (0.78-1.1) | 0.04 (0.03-0.05) | 0.48 (0.38-0.8) | 0.53 (0.35-1.4) | 0.08 (0.07-0.11) | 0.39 |
| **Ecuador** | | | | | | | | |
| UNFCCC | 0.55 | 0.39 | 0.09 | 0.02 | 0.04 | <0.01 | 0.01 | |
| Posterior | 0.57 (0.55-0.70) | 0.4 (0.39-0.48) | 0.1 (0.1-0.12) | 0.02 (0.02-0.02) | 0.04 (0.03-0.07) | 0.04 (0.03-0.07) | 0.01 (0.01-0.02) | 0.16 |
| **French Guiana** | | | | | | | | |
| Prior[d] | <0.01 | <0.01 | <0.01 | <0.01 | <0.01 | <0.01 | <0.01 | |
| Posterior | <0.01 | <0.01 | <0.01 | <0.01 | <0.01 | <0.01 | <0.01 | 0.052 |
| **Guyana** | | | | | | | | |
| UNFCCC | 0.05 | 0.02 | <0.01 | 0.02 | <0.01 | <0.01 | <0.01 | |
| Posterior | 0.07 (0.05-0.46) | 0.03 (0.02-0.13) | <0.01 | 0.03 (0.03-0.28) | <0.01 | <0.01 | <0.01 | 0.52 |
| **Paraguay** | | | | | | | | |
| UNFCCC | 0.86 | 0.75 | 0.06 | 0.02 | <0.01 | <0.01 | 0.03 | |
| Posterior | 0.93 (0.88-1) | 0.80 (0.76-0.86) | 0.06 (0.05-0.07) | 0.03 (0.03-0.03) | <0.01 | <0.01 | 0.04 (0.03-0.04) | 0.83 |
| **Peru** | | | | | | | | |
| UNFCCC | 1.1 | 0.46 | 0.44 | 0.05 | 0.09 | 0.01 | 0.04 | |
| Posterior | 2.4 (1.6-3.9) | 0.89 (0.64-1.5) | 1.1 (0.69-1.7) | 0.1 (0.07-0.19) | 0.22 (0.1-0.4) | 0.02 (0.01-0.03) | 0.07 (0.05-0.13) | 0.46 |
| **Suriname** | | | | | | | | |
| UNFCCC | 0.03 | <0.01 | <0.01 | 0.01 | 0.01 | <0.01 | <0.01 | |
| Posterior | 0.03 (0.03-0.04) | <0.01 | <0.01 | 0.01 (0.01-0.01) | 0.01 (<0.01-0.01) | 0.01 (<0.01-0.01) | <0.01 | 0.28 |
| **Uruguay** | | | | | | | | |
| UNFCCC | 0.77 | 0.7 | 0.05 | 0.01 | <0.01 | <0.01 | 0.01 | |
| Posterior | 1.1 (1.0-1.2) | 0.93 (0.8-1.0) | 0.14 (0.07-0.21) | 0.03 (0.03-0.03) | <0.01 | <0.01 | 0.02 (0.02-0.03) | 0.91 |
| **Venezuela** | | | | | | | | |
| UNFCCC | 2.5 | 0.89 | 0.25 | 0.02 | 1.3 | <0.01 | 0.02 | |
| Posterior | 7.0 (5.5-9.9) | 2.8 (2.0-4.5) | 0.67 (0.45-1.2) | 0.06 (0.04-0.10) | 3.4 (3.1-5.5) | <0.01 | 0.06 (0.04-0.12) | 0.68 |

[a] Prior estimates are from the latest country reports to the UNFCCC (see Sect. 2.2 for date). Posterior results are from the inversion of TROPOMI and GOSAT data for 2021 and are shown as the median of the inversion ensemble, with ranges from the inversion ensemble in parentheses.

[b] Minor sources including industry, stationary combustion, mobile combustion, aircraft, composting, and field burning of agricultural residues. These minor sources are taken from EDGAR v7.

[c] Ability of observations to quantify national anthropogenic emissions independently from the prior estimate (1 = fully, 0 = not at all) as measured by the diagonal terms of the averaging kernel matrix. Values are the median sensitivities across the inversion ensemble.

[d] There is no UNFCCC report for French Guiana, and our prior estimate is taken from a combination of GFEI v2 and EDGAR v7 (see text).





Waste (landfills and wastewater) is a large emissions sector across South America that sees a 50% upward correction to the
UNFCCC prior estimate. Countries estimate waste emissions using country-specific data on populations, waste generation
rates, and landfill monitoring along with IPCC defaults for methane yield (IPCC, 2019). Our waste estimate for Brazil is
consistent with its UNFCCC report but all other countries are found to be too low. Argentina (+59%) and Peru (+150%) see
the largest corrections. Peru lacks landfilling infrastructure so much of its waste resides in open dumpsites which may be
unaccounted for in UNFCCC reports (Ziegler-Rodriguez et al., 2019). Adding adequate closure systems to these dumpsites
could greatly reduce Peru's methane emissions (Cristóbal et al., 2022). In Argentina, TROPOMI has been used previously to
identify a strongly-emitting landfill in Buenos Aires (Maasakkers et al., 2022), where we also find an upward correction to
our prior estimate.

Figure 7 compares the prior and posterior oil/gas methane intensity for each country defined as the total emissions from the
oil/gas sector per unit of natural gas produced as methane (OGCI, 2022). We use national production data from EIA (EIA,
2023) and assume 90% of natural gas to be produced as methane as in Shen et al. (2023). We compare our posterior
intensities to those inferred from Shen et al. (2023) for individual countries worldwide and from Nathan et al. (2023) for
Venezuela in their inversions of TROPOMI data. Nathan et al. (2023) define methane intensity as the amount of methane
emitted per unit of combined oil and gas production, rather than just gas production. We find that Venezuela, Peru, and
Colombia have comparable posterior methane intensities to these previous studies, but Argentina's intensity is higher (5.9
(5.3-6.2) %) than inferred from Shen et al. (2023) (1.5%). Our large upward correction to emissions in Argentina is
concentrated near the San Jorge Gulf, Argentina's second largest oil field. All countries except Venezuela have methane
intensities of magnitudes comparable to the global average of 2.4% inferred by Shen et al. (2023) from inversion of
TROPOMI data and much higher than the industry target of 0.2% (OGCI, 2022), indicating a large potential to decrease
emissions. Venezuela has the highest posterior methane intensity (33 (29-54) %) in South America, which can be explained
by leakage from abandoned infrastructure as its oil production has declined (Nathan et al., 2023). Shen et al. (2023) found
Venezuela to have the highest methane intensity of any country globally.

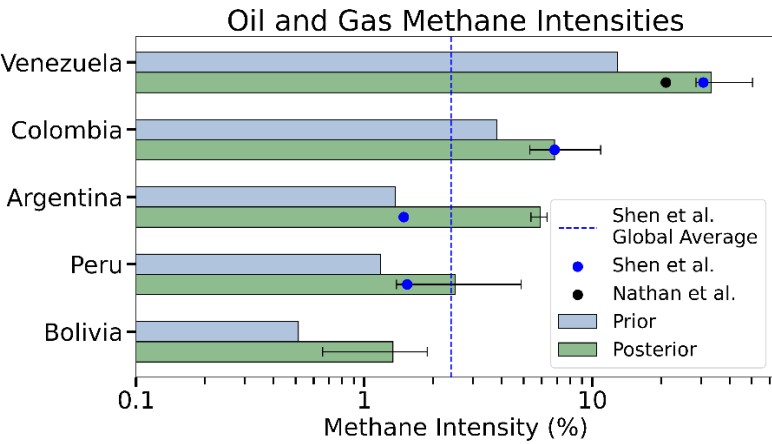



**Figure 7: Oil and gas methane intensities for major producing countries in South America.** The methane intensity is defined as the amount of methane emitted per unit of methane gas produced for our posterior result and that of Shen et al (2023). Nathan et al.
(2023) define methane intensity as the amount of methane emitted per unit of combined oil and gas production, rather than just gas production. Methane intensities computed from the prior and posterior emissions are compared to values inferred from Shen et al. (2023) in a previous inversion of TROPOMI data for May 2018-February 2020 (they only report an upper emission estimate of 1 Tg a$^{-1}$ for Bolivia) and from Nathan et al. (2023) in a TROPOMI inversion over Venezuela for 2019. Horizontal lines indicate the ranges from our inversion ensemble. The vertical line shows the global mean methane intensity of 2.4% reported by Shen et al.
430 (2023).

## 3.4 Livestock emissions

Livestock accounts for over 65% of anthropogenic methane emissions in South America (Table 1), and over 90% of this source is from enteric fermentation by cattle (FAOSTAT database, available at https://www.fao.org/faostat/en/#data, last accessed February 2024). Bottom-up inventories estimate emissions from enteric fermentation by multiplying cattle
populations by an emission factor per head. The emission factor depends on age, size, feed, cattle type, and environment. Dairy cattle and high milk productivity systems have higher emission factors due to higher dry matter intake and methane yield (Singaravadivelan et al., 2023). The IPCC (2019) gives different tiers of guidelines to incorporate this information into countries' bottom-up estimates. Tier 1 guidelines are to multiply cattle populations by emission factors that represent averages across all of Latin America. Tier 2 requires countries to calculate their own emission factors based on country-
specific data on feed, size, productivity, and amount of movement for different types of cattle. Tier 3 guidelines are not specific but could include the development of sophisticated models considering diet composition or the fermentation process in more detail (Bannink et al., 2011).

Tier 1 emission factors for Latin America are calculated by the IPCC (2019) using data from 52 publications of which 32 are for Brazil. These emission factors are 58 and 55 kg CH$_4$ head$^{-1}$ a$^{-1}$ for non-dairy cattle and 78 and 103 kg CH$_4$ head$^{-1}$ a$^{-1}$ for
dairy cattle in low and high productivity systems, respectively. Because these values are presented as averages across Latin America, countries for which livestock is a dominant emission source are encouraged by IPCC (2019) to use Tier 2 or Tier 3 methods instead.

Many South American countries describe using a combination of Tier 2 and Tier 1 methods in their UNFCCC reports, often with little detail. Countries report a total amount of emissions from enteric fermentation, so here we infer mean bottom-up
emission factors for each country by multiplying this reported number by the proportion from cattle estimated by FAO (Table 3) and dividing by the total number of cattle from FAO for the reported year. We find that all the top livestock emitters (Argentina, Brazil, Bolivia, Colombia, Peru, and Venezuela) have inferred emission factors close to the IPCC Tier 1 emission factors for non-dairy cattle, with emission factors ranging from 46 to 60 kg CH$_4$ head$^{-1}$ a$^{-1}$ (Figure 8 and Table 3).



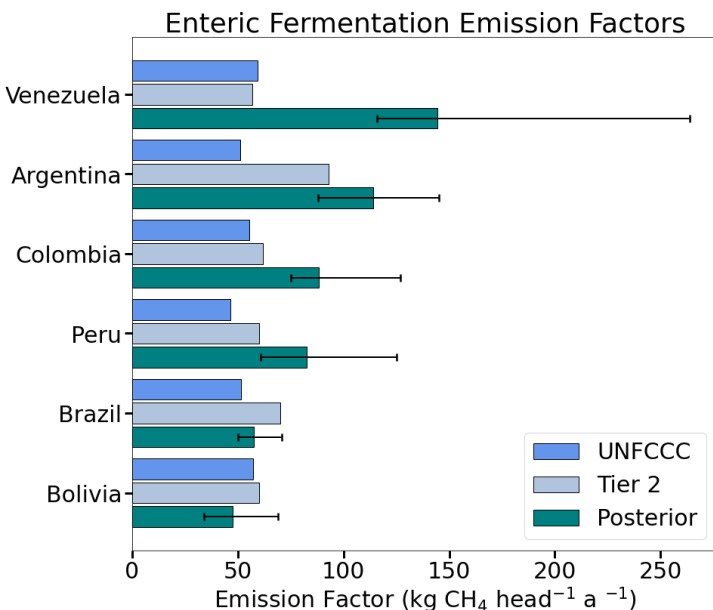

**Figure 8: Mean cattle enteric fermentation emission factors by country. Posterior and UNFCCC emission factors are derived by taking each source's estimate for enteric fermentation from livestock, multiplying it by the fraction of livestock emissions from cattle enteric fermentation (Table 3), and dividing by the number of cattle estimated from FAOSTAT (FAOSTAT database, available at https://www.fao.org/faostat/en/#data, last accessed February 2024). Tier 2 emission factors are calculated using country-specific data on cattle weight. More details on these calculations can be found in Table 3. Horizontal lines show the ranges of posterior estimates from our inversion ensemble.**

We obtain a posterior estimate of total enteric fermentation emissions from livestock for each country in our inversion. Manure management and enteric fermentation are highly correlated with each other, but because manure management emissions are comparatively small, we consider our enteric fermentation estimate to be sufficiently accurate for the following analysis. Because this total can also include enteric fermentation from other animals which can be significant in some countries (Table 3), we multiply the total enteric fermentation by the proportion from cattle estimated by FAO. We calculate posterior emission factors for each country by dividing these estimates by FAO cattle populations in 2021. We find that emission factors are larger than those inferred from the UNFCCC reports except for Brazil and Bolivia (Figure 8).

Table 3 lists additional bottom-up emission factor estimates from Climate TRACE (Climate TRACE Emissions Inventory, 2022) and EDGARv7 (Crippa et al., 2022). Climate TRACE is a bottom-up inventory that uses 2006 IPCC emission factors with FAOSTAT cattle populations to calculate country totals for enteric fermentation methane, separating cattle by dairy, feedlot, and pasture (Climate TRACE Emissions Inventory, 2022). EDGAR also uses IPCC methods with additional country-specific information on animal body weights. We derive emission factors by taking country enteric fermentation emission totals from each inventory, multiplying by the proportion from cattle estimated by FAO, and dividing by FAO cattle populations in 2021. Both are within 20% of the UNFCCC-reported values.



**Table 3: Mean livestock emission factors by country[a]**

| Country | Argentina | Brazil | Bolivia | Colombia | Peru | Venezuela |
|---|---|---|---|---|---|---|
| **Cattle Weight (kg)[b]** | 430 | 659 | 371 | 430 | 273 | 464 |
| **Milk Production (kg head$^{-1}$ a$^{-1}$)[c]** | 7570 | 2280 | 2490 | 1900 | 2340 | 939 |
| **Fraction from cattle[d]** | 0.94 | 0.95 | 0.76 | 0.93 | 0.55 | 0.94 |
| **Emission Factor (kg head$^{-1}$ a$^{-1}$)[e]** | | | | | | |
| **Our work (posterior)** | 114 (88 - 145) | 58 (50 - 71) | 48 (34 - 69) | 88 (75 - 127) | 83 (61 - 125) | 145 (116 - 264) |
| **UNFCCC** | 51 | 52 | 57 | 56 | 46 | 60 |
| **Tier 2[f]** | 93 | 70 | 60 | 62 | 60 | 57 |
| **EDGAR** | 64 | 65 | 51 | 60 | 39 | 63 |
| **Climate TRACE** | 58 | 55 | 54 | 55 | 58 | 60 |
| **Benaouda et al.[g]** | 69 | 106 | 60 | 69 | 44 | 75 |

[a] Countries with averaging kernel sensitivities for livestock less than 0.2 or with a high correlation between livestock and other sectors are not shown.

[b] Cattle weights are calculated using chilled carcass weights from FAOSTAT (FAOSTAT database, available at https://www.fao.org/faostat/en/#data, last accessed February 2024), assuming a dressing percentage of 55% and a chill shrinkage of 3%.

[c] Milk production data are from FAOSTAT.

[d] Denotes the fraction of livestock emissions from cattle enteric fermentation, calculated from FAOSTAT methane emissions estimates.

[e] Emission factors from our work (posterior), UNFCCC, EDGAR, and Climate TRACE are calculated by taking each source's estimate for enteric fermentation from livestock, multiplying it by the fraction of livestock emissions from cattle enteric fermentation[d] and dividing by the number of cattle estimated from FAOSTAT.

[f] Tier 2 estimates are calculated using equation 10.21 in Chapter 10 of the IPCC's 2019 refinement to the 2006 IPCC Guidelines for

National Greenhouse Gas Inventories (IPCC, 2019). We calculate emission factors for each cattle type and average these together, weighting by population of each cattle type, to obtain a mean emission factor for each country. County-specific cattle body weights[b] are used for mature cattle. Country-specific milk production[c] is used to calculate the net energy from lactation (equation 10.8, IPCC, 2019) which is incorporated into the dairy cattle emission factor. Elsewhere Latin America default values are used (tables 10A.1, 10A.2, IPCC, 2019).

[g] These estimates are calculated using model 1.1 from Benaouda et al. (2020) and country-specific cattle body weights[b].

We also calculate bottom-up emission factors using IPCC Tier 2 methods, incorporating country-specific data from FAO on milk production and cattle body weights but using Latin America defaults elsewhere. We find that our Tier 2 estimate aligns better to our posterior estimate in Argentina. This is due to a strong dependence of Tier 2 estimates on milk production;

Argentina has the highest milk production per cow of all South American countries (Table 3). Benaouda et al. (2020) and Congio et al. (2023) both offer cattle weight-dependent linear equations to predict emission factors based on empirical data



from Latin American countries, but the Congio et al. prediction does not capture variation across countries (emission factors range from 92-95 kg $CH_4$ head$^{-1}$ a$^{-1}$). Even with a stronger dependence on body weight, the Benaouda et al. prediction reflects that cattle weight alone cannot predict the variations in emission factors we see in our posterior estimate: Brazil's emission factor is too high (106 kg $CH_4$ head$^{-1}$ a$^{-1}$) and all other countries' emission factors except Bolivia are too low.

Overall, we find from the inversion of TROPOMI and GOSAT data a much larger range of mean emission factors from enteric fermentation by cattle in individual countries (48-145 kg $CH_4$ head$^{-1}$ a$^{-1}$) than inferred from UNFCC reports (46-60 kg $CH_4$ head$^{-1}$ a$^{-1}$). The countries at the low end of our range (Brazil and Bolivia) are consistent with UNFCCC reports but other countries are much higher. The difference with Argentina would be reconciled if the UNFCCC report used IPCC Tier 2 methods to account for larger emissions from dairy cattle. The differences for Colombia, Peru, and Venezuela are not as readily explained. Salas-Riega et al. (2022) showed that enteric emissions for both lactating and non-lactating cattle in the Peruvian high Andes were higher than derived from IPCC Tier 2 methods (119 and 97 kg $CH_4$ head$^{-1}$ a$^{-1}$ for lactating and non-lactating cattle, respectively). Benaouda et al. (2020) reviewed daily measurements of cattle enteric fermentation in Latin America and found a wide range of emission factors, from 18 to 239 kg $CH_4$ head$^{-1}$ a$^{-1}$ with an average of 68 kg $CH_4$ head$^{-1}$ a$^{-1}$. Our posterior emission factors could also be influenced by uncertainties in cattle populations or emissions from manure and other animals.

One possible weakness in our inversion is the reliance on EDGAR v7 for the prior spatial distribution of livestock emissions on the 0.25°×0.3125° grid. EDGAR spatially allocates emissions by using an array of proxy datasets including FAOSTAT animal density, global land cover data, and in-house proxy data that is not publicly available. Errors in this spatial distribution would propagate to inversion results by affecting both the optimal solution to the inverse problem (Yu et al., 2022) and the attribution of 0.25°×0.3125° posterior emissions to specific sectors (Shen et al., 2021). Figure 9 shows 2021 emissions from 779 individual feedlots and dairies in Northern Argentina and Southern Brazil estimated by Climate TRACE by using artificial intelligence to identify facility locations in PlanetScope (Planet Team, 2021) satellite imagery, assuming livestock numbers to be proportional to facility area, and applying 2006 IPCC emission factors (Davitt et al., 2023). The high emissions in northern Argentina do not match the spatial distribution from EDGAR (Figure 2). The right panel of Figure 9 compares our prior and posterior emission estimates to the Climate TRACE values for inversion grid cells dominated by livestock. Our values are higher because Climate TRACE estimates are limited to larger feedlots visible in PlanetScope imagery. However, we find better spatial correlations between Climate TRACE and our posterior emissions ($r = 0.44$, $p = 0.0004$) than our prior emissions ($r = -0.11$, $p = .42$). The Climate TRACE database could be useful as prior estimate for future inversions but would need to be more comprehensive.



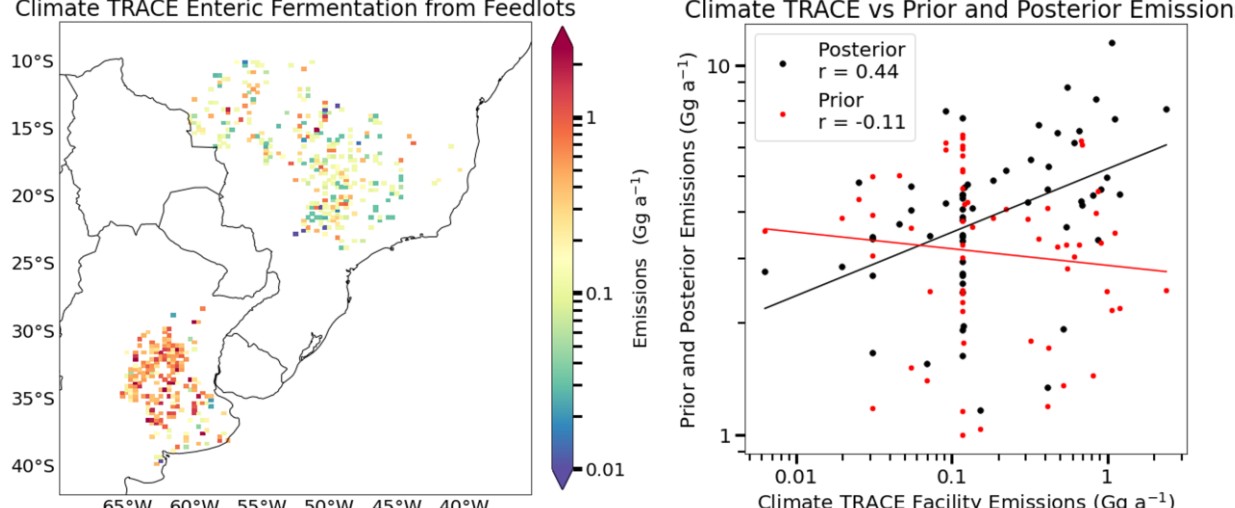

**Figure 9:** Satellite-informed spatial distribution of enteric fermentation emissions from feedlots and dairies in southern Brazil and northern Argentina from Climate TRACE (Davitt et al., 2023). The left panel shows the distribution of emissions from individual feedlots averaged onto the native 0.25°×0.3125° grid cells of our inversion. The right panel compares our prior and posterior emissions to the Climate TRACE data for grid cells where the inversion has averaging kernel sensitivities greater than 0.5, fewer than 50 grid cells aggregated within the state vector element, and more than 50% of prior emissions are from livestock. Reduced-major-axis linear regressions are also shown.

## 4 Conclusions

We used 2021 TROPOMI and GOSAT satellite observations of atmospheric methane ($XCH_4$) in a high-resolution analytical inversion to infer methane emissions from South America at up to $25\,km \times 25\,km$ resolution with an emphasis on the contributions from anthropogenic emissions in individual countries. The goal of this work was to evaluate the national inventories submitted to the United Nations Framework Convention on Climate Change (UNFCCC) under the Paris Agreement and to identify opportunities to improve countries' bottom-up reporting methods.

We used national emissions inventories reported by individual countries to the UNFCCC, gridded using EDGARv7, as the prior estimate in the inversion. For wetlands we used two alternative prior estimates, from WetCHARTs and LPJ-MERRA2, with different spatial distributions. We obtained best posterior estimates of emissions analytically through Bayesian synthesis of these prior estimates with the information from the TROPOMI and GOSAT observations. We used a blended TROPOMI+GOSAT product that corrects spatially variable biases and artifacts in the TROPOMI data using information from GOSAT. Although TROPOMI data is in general much denser than GOSAT, GOSAT provides unique coverage over the Amazon where TROPOMI data is sparse. The inversion used variable resolution with a Gaussian mixture model (GMM) state vector that enforces native 25-km resolution in source regions with high observation density. Analytical solution to the inversion enabled the creation of an inversion ensemble with 54 members for conservative uncertainty estimates on posterior emissions.





Total posterior emissions for South America are 121 (109-137) Tg a⁻¹, where the best estimate is the median of our inversion ensemble and the range is in parentheses. This is significantly higher than the prior estimate of 96 Tg a⁻¹. Most of the increase is from anthropogenic emissions, which increase from 31 Tg a⁻¹ in the UNFCCC reports used as prior estimates to 48 (41-56) Tg a⁻¹. Anthropogenic emissions are dominated by livestock (65%), followed by waste (16%), and oil/gas (13%). Total anthropogenic emissions in South America are 55% higher than in the prior estimate, reflecting increases in emissions from oil/gas (+158%), livestock (+48%), and waste (+37%).

We obtain best posterior estimates of wetland emissions from the Amazon (32 (29-44) Tg a⁻¹), the Bolivian Amazon (2.8 (1.6-4.4) Tg a⁻¹), the Pantanal (1.5 (1.2-1.8) Tg a⁻¹), and the Paraná (2.0 (1.8-2.2) Tg a⁻¹). Our estimate for the Amazon is consistent with past estimates, but our estimate for the Pantanal is lower. Emissions from the Paraná are much higher than in either WetCHARTS or LPJ-MERRA2. Posterior wetland continental total emissions agree better with LPJ-MERRA2 than WetCHARTs, but the posterior spatial distribution better matches WetCHARTs.

We compare the UNFCCC reports of anthropogenic emissions from individual countries to our best sector-resolved posterior estimates. We find that TROPOMI and GOSAT observations can effectively resolve emissions from individual countries except Ecuador and Suriname. The top seven emitting countries including Brazil, Argentina, Venezuela, Colombia, Peru, Bolivia, and Paraguay make up 93% of the total anthropogenic emissions in the region, with Brazil contributing the highest amount (40%). All countries except Bolivia, Brazil, and Suriname show significant upward corrections to their UNFCCC-reported anthropogenic emissions. Waste emissions are underestimated in the UNFCCC reports, particularly in Peru. Oil/gas emissions are underestimated in all producing countries except Brazil. We find high methane intensities from the oil/gas sector in Venezuela (33 (29-54) %), Colombia (6.5 (5.1-10.8) %), and Argentina (5.9 (5.3-6.2) %).

We examined livestock emissions and their reporting to UNFCCC in more detail. These emissions are over 90% from enteric fermentation by cattle. The average emission factors per head of cattle in the UNFCCC reports range from 46 to 60 kg CH₄ head⁻¹ a⁻¹, close to the IPCC Tier 1 emission factors for non-dairy cattle and consistent with other bottom-up estimates from the EDGAR and Climate TRACE inventories. Aside from Brazil and Bolivia, we find these emission factors to be a factor of 2 too low. Better accounting for dairy cattle emissions through IPCC Tier 2 estimates corrects the discrepancy for Argentina but not for other countries.

## 5 Data Availability

The blended TROPOMI+GOSAT satellite observations version 2 are available at https://registry.opendata.aws/blended-tropomi-gosat-methane (Balasus et al., 2023). The GOSAT methane retrievals version 9.0 are available at https://doi.org/10.5285/18ef8247f52a4cb6a14013f8235cc1eb (Parker and Boesch, 2020). Oil, gas, and coal emissions from the GFEIv2 inventory are available at https://doi.org/10.7910/DVN/HH4EUM (Scarpelli and Jacob, 2021). Methane emissions by sector from EDGARv7 are available at https://edgar.jrc.ec.europa.eu/dataset_ghg70 (Crippa et al., 2022). Wetland emissions from WetCHARTs v1.3.1 are available at https://doi.org/10.3334/ORNLDAAC/1915 (Ma et al., 2021).



## 6 Author contributions

SH and DJJ contributed to the study conceptualization. SH conducted the data and modeling analysis with contributions from DJJ, ZC, HN, AD, DJV, MPS, NB, LAE, JDE, EP, CAR, JW, IA, RJP, and JDM. SH and DJJ wrote the paper with 590 input from all authors.

## 7 Competing Interests

At least one of the co-authors is a member of the editorial board of Atmospheric Chemistry and Physics.

## 8 Acknowledgements

This work was supported by the Harvard University Climate Change Solutions Fund (CCSF), by the NASA Carbon 595 Monitoring System (CMS), and by a National Science Foundation Graduate Research Fellowship under grant no. DGE 2140743. This research has also been funded in the framework of UNEP's International Methane Emissions Observatory (IMEO). Part of this research was carried out at the Jet Propulsion Laboratory, California Institute of Technology, under a contract with the National Aeronautics and Space Administration supported by NASA ROSES Grant 18-CMS18-0018. This research was also supported in part by an appointment to the NASA Postdoctoral Program at the Jet Propulsion Laboratory, 600 California Institute of Technology, administered by Oak Ridge Associated Universities under contract with NASA. RJP is funded via the UK National Centre for Earth Observation (Grant: NE/W004895/1). The GOSAT data generation was supported by the Natural Environment Research Council (NERC grant reference number NE/X019071/1, "UK EO Climate Information Service"). We also acknowledge funding from the ESA GHG-CCI and Copernicus C3S projects (grant no. C3S2_312a_Lot2). This research used the ALICE high-performance computing facility at the University of Leicester for the 605 GOSAT retrievals. We thank the Japanese Aerospace Exploration Agency, National Institute for Environmental Studies and the Ministry of Environment for the GOSAT data and their continuous support as part of the Joint Research Agreement.

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
