# Peer review of "Satellite quantification of methane emissions from South American countries: A high-resolution inversion of TROPOMI and GOSAT observations"

_EGUsphere, 2024_

## Author Comment (AC1)

We thank the South American researchers for their detailed suggestions and comments on the manuscript. Please see below our responses to the comments. We have listed out the community comments in black and the replies in blue.

**Community Comment #1**

María Cazorla[1], Néstor Rojas[2], Sebastián Diez[3], Nicolás Huneeus[4], Dara Salcedo[5], Rafael Fernández[6], Laura Dawidowski [7], Eduardo Luccini[8], Gerardo Carbajal Benítez[9]
[1]Universidad San Francisco de Quito USFQ, Ecuador
[2]Universidad Nacional de Colombia, Colombia
[3]Universidad del Desarrollo, Chile
[4]Universidad de Chile, Chile
[5]Universidad Nacional Autónoma de México, México
[6] Universidad Nacional de Cuyo, Argentina
[7] Comisión Nacional de Energía Atómica, Argentina
[8]National Scientific and Technical Research Council of Argentina (CONICET–CEPROCOR), Argentina
[9]Servicio Meteorológico Nacional, Argentina

The paper by Hanock et. al. presents results from a top-down inversion model of methane emissions by economy sector for every country in South America. The authors use as input satellite products from TROPOMI and GOSAT, and the EDGARv7 inventory (with WetCHARTs and LPJ-Merra2 for wetlands) to spatially distribute emissions from livestock, waste, and rice production. As main findings, the authors highlight discrepancies with national anthropogenic emission inventories reported by many South American countries to the UNFCCC and propose recalculated values. In this regard, we have a series of comments that we think should be addressed:

**Major comments:**

1. The paper deals with matters that are very specific to individual countries and economy sectors in South America. However, the authors did not look for the insight and knowledge of South American scientists to contrast their results, enhance the discussion, and find explanations to their findings that reflect the reality of each country. We believe studies of this nature require local knowledge and collaboration, beyond the mere citation of some papers, to improve the credibility of the results and to correct potential biases. This critical aspect is missing in this paper. Including local expertise is crucial to identifying country-specific factors influencing emissions that can be easily overlooked by external researchers. As a good scientific practice, it is important that scientists in the Northern Hemisphere acknowledge the fact that every country in South America has experts who can be consulted and invited to contribute as co-authors to ensure studies about this region, especially those that deal with sensitive topics such as GHG emissions, are not done using a one-sided perspective.

Thank you for your interest in our work and thoughtful comments. Your stated need for input from local researchers, especially those with expertise in bottom-up calculations of methane emissions, is very well taken and we couldn't agree more. A nagging problem in methane research is the lack of communication between top-down (TD) and bottom-up (BU) scientists. It is not easy to get BU scientists interested in TD work! And yet the TD work gets all its meaning if it can help inform the BU inventories. Your interest is thrilling to us and we very much want to engage with you in future work. We will communicate on this by separate email. In the meantime, we are eager to include your concerns in the revised draft and include you as co-authors, either on this manuscript or on follow-up studies for specific countries in which we would work in partnership with you.

The present work is intended to be a starting point for addressing South American methane with satellite observations. Our hope is that this work can be followed up by country-specific studies working with

local governments and scientists. We are in fact doing precisely this right now by conducting a higher-resolution inversion over Colombia in collaboration with local scientists and with the goal of making a better connection between TD and BU estimates – and how TD information can help BU inventories. We would like to do this for your individual countries as well.

2. The paper implies that the results from the satellite inversion are correct, and the bottom-up inventories are not. Particularly, this is evident from the use of the terms "correct or corrections" throughout the paper when referring to national inventories and reports. Thus, the authors of the paper assume that there is no error on satellite retrievals, but the error is in the UNFCCC reports. These are major statements that need to be demonstrated. One possible way is to compare the inversion results with another approach to estimate $CH_4$ emissions or to consult specific information on how national inventories were produced to point out to specific issues.

The paper does not intend to imply that results from the inversion are correct. It is common in the TD literature to refer to 'correction' and correct (verb) as the change from the BU prior. This does not mean that the TD results are correct! We will edit the text to avoid such misinterpretation. We take into account the errors in the satellite retrieval as explained in Section 2.3 (lines 261-276). These errors are fairly well characterized. We are not the first TD study to find that BU inventories underestimate methane emissions in South America (Worden et al., 2022). Our purpose is to demonstrate this underestimate at higher resolution and begin to explore potential causes. Our inversion uses countries' UNFCCC sectoral emission totals as prior in the inversion, so we interpret our results as an adjustment to those inventories based on information from the satellite data. We will do better in revision in discussing potential biases.

3. The authors' approach to assessing the emissions that countries report to the UNFCCC does not coincide with the approach that the countries use in their reporting. This situation produced a very significant bias in the results presented in this work. To explain how biases come about, we will focus on the authors' assessment of livestock emissions, whose $CH_4$ emissions are a key category for almost all countries in the region. Just as an example, we will focus on emissions from Argentina:

- The authors seem to imply that countries' reporting to the UNFCCC is not very transparent. However, the original reports are in Spanish, contain great level of detail and are publicly available. For example, Argentina in its reports (https://unfccc.int/sites/default/files/resource/argentina-bur5.pdf) uses 238 pages (from 536 to 774) to thoroughly explain the emissions of this category. The country applies a level 2 approach and presents the enormous variability of the type of livestock considered, as well as some different modal systems (for winter and summer) that take place in the country.

- Despite the significant level of detail in the information provided by the country, the authors only use the total emissions reported for this category.

- All the complexity of the sector is simplified by taking average variables published by FAO, which is well known not to represent the sector in Argentina (and in many of the countries presented in the manuscript).

- With the FAO information, authors estimate emission factors, which they call "from UNFCCC", when they actually differ substantially from what the country reports to the UNFCCC.

In the conclusions the authors state: "We compare the UNFCCC reports of anthropogenic emissions from individual countries to our best sector-resolved posterior estimates". However, authors do not compare the UNFCCC reports from individual countries but compare their own estimates with those obtained in their calculations from satellite information.

Thank you for raising this point. We state that "Many South American countries describe using a combination of Tier 2 and Tier 1 methods in their UNFCCC reports, often with little detail," which devalues countries like Argentina that have in-depth bottom-up reporting methods for livestock. We will correct this language in our revised manuscript. Because not all countries provide such detail in their UNFCCC reports, we used the greatest common denominator of complexity in our analysis which is the emission totals by sector.

We take countries' totals for enteric fermentation methane emissions from the most recent UNFCCC reports (detailed in lines 140-143) and divide them by the number of cattle estimated by FAO to get an implied national emission factor. We agree that this may not be the emission factor that countries use in their UNFCCC reports. Our intention is not to offer recalculated emission factors, but to point out the mismatch between the bottom-up and top-down estimates and begin to explore the potential causes. In our revised manuscript, we will instead compare country totals for enteric fermentation directly obtained from UNFCCC reports with our Tier 2 and top-down posterior estimates rather than emission factors. We will ensure also that impact of sources of uncertainty such as the prior emissions distribution is emphasized in the text.

4. The paper lacks context in the sense that methane emissions in South America, even with the proposed recalculation, are not compared against emissions of the main global emitters. In particular, the authors emphasize differences found between totals reported by countries and calculated using the satellite inversion method by sectors. To provide a fair and comprehensive context, the authors should compare South American emissions, both reported and recalculated as well as the difference, against those from major emitting countries worldwide. Such comparison should include the uncertainties in the satellite products (see the following comment). Without this comparison, the results can be misleading and fail to convey the true significance of the findings.
Thank you for this suggestion. We will add additional text to interpret our results in the context of top methane emitters globally.

5. Validation of satellite retrievals against in situ measurements is crucial to ensure data reliability. However, there are no systematic validations of satellite products in South America. TROPOMI and GOSAT methane products (as well as other satellite products) heavily rely on validations mostly in the Northern Hemisphere. Consequently, there should be a quantification of the uncertainty in the results due to this regional validation gap.
Thank you for this excellent point. We will add language to emphasize this uncertainty in the revised manuscript.

**Specific comments:**

- The manuscript states that the global retrieval success rate for GOSAT is 23.5%. It would be beneficial to include the retrieval success rate specifically for South America, including differences between tropical and subtropical areas.

Great point. We will address this in revision.

- Line 97 states that "We also subtract 9.2 ppb from all GOSAT observations following Balasus et al. (2023) to remove the global mean bias versus TCCON". Is this valid for South America? There are probably large variations between the Northern and Southern Hemispheres. Some regions or periods might be overcorrected or undercorrected, potentially affecting the spatial and temporal accuracy of the emission estimates. Since the TCCON network lacks measurements in South America, it would be important to perform cross-validations with other independent measurements (e.g., aircraft data, in situ measurements) to verify the applied correction.

Additionally, conducting sensitivity analyses to assess how the inversion results vary with different correction values would help understand the impact of this assumption.

Balasus et al. (2023) use GOSAT observations calibrated to a global mean bias of 0 ppb relative to TCCON to construct their blended TROPOMI+GOSAT product. We apply the same 9.2 downward correction to all GOSAT observations to enforce consistency between the blended TROPOMI product and the GOSAT observations we use in our inversion. The global bias relative to TCCON is unimportant for regional inversions because it is effectively corrected through the boundary conditions.

- Figure 2: This figure seems to be based on global inventories merged with data from different countries. It would be appropriate to compare with national/regional inventories. Examples for Argentina are provided below, but an exhaustive review should be done by authors for all countries in South America.
  https://isprs-annals.copernicus.org/articles/IV-3-W2-2020/107/2020/
  https://www.sciencedirect.com/science/article/pii/S1352231019308866

Figure 2 depicts the prior estimates used in our inversion. Anthropogenic emissions for this figure are taken from countries' UNFCCC reports. The UNFCCC totals from each sector are spatially distributed using global inventories GFEIv2 for fuel sources and EDGARv7 for livestock, waste, and rice. We agree that it would be ideal to instead use national/regional inventories for all countries as the prior spatial distribution for our inversion. To the best of our knowledge, such spatially gridded inventories are not available for all South American countries, and including the existing national inventories such as Argentina's as prior in the inversion would entail redoing all of the 0.25° × 0.3125° perturbation simulations, representing over 100 thousand hours of computation time. This is beyond the scope of this manuscript which is intended to serve as a first look at a high-resolution, top-down estimate of methane emissions over South America. We hope that future work can obtain higher-resolution results over specific countries and we are eager to work with local scientists to interpret top-down results in the context of spatially gridded national inventories.

- Figure 3: Posterior emission map (b) follows the shape and intensity of the prior emission map (a). This highlights the importance of using the best emission estimate as an a-priori starting point.

The log scale makes it difficult to see the differences in patterns between (a) and (b). More useful is the posterior/prior ratio in panel (c). We will make that point in revision.

- Lines 75 to 77: "Livestock emissions are underestimated in all four of these countries. Argentina and Venezuela also underestimate their oil/gas emissions." Why? Any explanation?

We further analyze the livestock sector in section 3.4 and the oil/gas sector in lines 408-421. We relate the livestock sector underestimate to emission factors for enteric fermentation and the oil/gas underestimate to methane intensity. We look forward to being able to dig down deeper in future work.

- Figure 6: Why are wetland emissions not shown here? It is evident that the largest discrepancies between UNFCCC and Posterior are Livestock and Oil/Gas, followed by Waste, but no discussion about them is given.

We report emissions for wetlands in section 3.2 instead to distinguish between natural and anthropogenic emissions, since discrepancies between prior and posterior wetland emissions would be due to differences in the wetland prior inventories we use rather than countries' UNFCCC reports.

- The paper mentions that "There are few observations over the mountainous Andes, affecting much of Chile and Peru, , so that the inversion for those countries relies significantly on glint observations offshore and on observations of transported methane.". How does this impact the uncertainties for these countries? We suggest expanding on this point to discuss the implications for the accuracy and reliability of the inversion results in these regions.

The smaller number of observations over these countries is reflected in their lower averaging kernel sensitivities shown in Table 2. We will emphasize this point in the revised draft.

---

## Author Response (AR1)

We thank the South American researchers for their feedback on this manuscript. We addressed their specific comments in a separate reply, but we have also now added María Cazorla, Laura Dawidowski, and Sebastián Diez as co-authors. Their assistance in the revision process has helped strengthen the manuscript and add a much-needed local perspective, particularly with regard to our interpretation of inversion results.

We also thank the reviewers for their suggestions and comments on the manuscript. Please see below our responses to the comments. We have listed out the reviewer comments in black and the replies in blue.

**Reviewer #1**

Hancock et al. present a comprehensive study that integrates satellite observations of $CH_4$ with an atmospheric transport model and prior emissions estimates to derive an optimal set of methane emissions for South American countries. This analysis further disaggregates the optimized emissions by sector and country, enhancing the atmospheric constraint on South American methane emissions. The topic is significant, and well-aligned with the scope of ACP. I am not an expert on inverse modeling but rather on in-situ measurements so I can't comment deeply on the mathematical aspects of the inversion method and learnt a lot. Below, I outline a few questions and suggestions.

**Line 93:** what is the retrieval success rate specifically for South America?

The retrieval success rate for South America is 28.4%. We now note this in the text (lines 99-100).

**Regarding Fig. 1:** How would the seasonal variability of methane concentration affect these emission estimates? A spatial plot of the total number of samplings in each season during 2021 might be useful to include in the supplement. Is it possible to extend this inverse modeling setup to estimate total methane emissions on a monthly scale?

Thank you for this suggestion. Since we have higher observation density in the dry season (June-September), our emissions estimates are more heavily based on observations during this period. We have now included a figure in the supplement showing seasonal TROPOMI and GOSAT observation density (Figure S1). We also show in Figure S1 that the mean bias between GEOS-Chem and TROPOMI+GOSAT $XCH_4$ is lower in the posterior than in the prior in all seasons.

It is possible to use a similar inverse modeling setup to estimate methane emissions on a monthly scale. Varon et al. (2023) demonstrated this over the Permian Basin with weekly monitoring of emissions using TROPOMI. While looking at seasonal variability over South America would be very interesting, it would require redoing all of the $0.25° \times 0.3125°$ perturbation simulations, representing over 100 thousand hours of computation time. We hope this can be a focus of future work.

**Line 115:** "There are few observations over the mountainous Andes, affecting much of Chile and Peru, so the inversion for those countries relies significantly on glint observations offshore and on observations of transported methane." How does this affect the uncertainties in estimating emissions for this area?

The smaller number of observations over these countries is reflected in their lower averaging kernel sensitivities shown in Table 2. We now emphasize this in the text (lines 420-421).

How does the inverse model handle temporal variability in emissions? While the model optimizes the overall magnitude and spatial patterns in emissions, does it also optimize seasonal or year-to-year variability? In other words, does the model assume that the temporal distribution of emissions is known or fixed according to the prior temporal distribution?

We optimize annual emissions such that the inversion assumes the temporal distribution of emissions follows that of the prior. The primary driver of seasonality in emissions is wetlands, and we test two different wetland prior inventories to characterize the uncertainty that comes from this, but we do not optimize the seasonality of emissions.

**Line 206:** "We use 600 Gaussian functions as state vector elements to balance aggregation and smoothing errors." While a reference is provided, a brief explanation of why 600 Gaussian functions are used would be helpful.

Thank you. We have added this to the manuscript (lines 223-225).

**Regarding Fig. 3**: The ratios of the posterior/prior emissions in Fig. 3c show values close to zero or over 2 in many areas (e.g., Bolivia and Argentina). Does this imply that the inversion zeroed or doubled emissions? If so, are the resulting emissions reasonable?

Thank you for this question. Yes, the ratios indicate what the prior emissions in a particular grid cell should be multiplied by to obtain the posterior emissions, so a ratio of 2 would mean the emissions are doubled and a ratio of 0.5 would mean the emissions are halved. We consider the resulting emissions to be reasonable, especially because the very large and very small ratios are generally limited to areas with low prior emissions and thus the magnitude of change in emissions is small.

Given the goal of the paper is to "evaluate the national inventories submitted to the United Nations Framework Convention on Climate Change (UNFCCC) under the Paris Agreement and to identify opportunities to improve countries' bottom-up reporting methods," including model-data comparisons against independent $CH_4$ observations is crucial for evaluating the inverse model. While there is a comparison with aircraft measurements in the Bolivian Amazon region, this is not enough. A supplement showing the bias of methane concentration in the prior and posterior run relative to in-situ observations in South America would be very informative.

Thank you for this suggestion. We have now included a comparison with the Amazon Tall Tower Observatory (ATTO) in Brazil (Figure 5).

**Minor Point: Line 98:** TCCON is first mentioned here. It should be referred to as the "Total Carbon Column Observing Network (TCCON)." Additionally, providing a brief explanation of TCCON would help readers unfamiliar with this field understand its purpose.

Thank you for this suggestion. We have added this to the manuscript (lines 104-105).

**Reviewer #2**

This paper presents a comprehensive analysis of methane emissions across South America. By employing high-resolution satellite data from TROPOMI and GOSAT, the authors present a comprehensive and spatially detailed estimation of methane sources, mainly anthropogenic sources, which constitutes a significant contribution to the comprehension of regional methane budgets. However, there are a few aspects of the paper that could be addressed to yield a more robust and comprehensive analysis.

The integration of data from two complementary satellite instruments helps to improve estimates through the use of inverse modelling. Nevertheless, a validation with independent CH4 observations available in South America would be beneficial for this study. For instance, the authors could undertake a comparison of the posterior mole fraction with data obtained from the ATTO tower in Brazil and vertical profiles in the Amazon region.

Thank you for the suggestion. We have now included a comparison with the ATTO tower in Brazil (Figure 5).

With regard to the regional budget, it would be beneficial to include a comparison with other previously published top-down estimates of total methane emissions for each country, such as those included in the Global Methane Budget.

The authors could additionally provide insight into the implications of their findings for policymaking strategies. This would be beneficial, as they evaluated national anthropogenic emissions inventories reported by individual countries to the UNFCCC. In particular, the authors could elucidate how these data could help local governments to mitigate methane emissions.

Thank you for this suggestion. We have included a regional comparison to estimates from the Global Methane Budget (Saunois et al. (2024)) and a global inversion by Worden et al. (2022) in the supplement (Figure S2).

We have changed the language of the paper to clarify that the intention of this work is not to evaluate countries' UNFCCC reports, but rather to begin to explore possible causes of the mismatch between top-down methane emissions estimates and bottom-up inventories that have been identified in global inversions (Worden et al., 2022). This work is only a starting point for addressing South American methane with satellite observations and our hope is that it can be followed up by country-specific studies working with local governments and scientists that will provide more policy-relevant results.

Specific comments are provided below.
Line 20: The term "correcting" may suggest that top-down estimates are inherently more accurate than bottom-up estimates, whereas both approaches are subject to their own sets of uncertainties. To prevent any potential misinterpretations, it would be helpful to use a term such as "adjusting" or "reconciling" when discussing the comparison or combination of these estimates. It is also important to discuss the limitations of both methods, top-down and bottom-up estimates, in the paper.
Thank you for this suggestion. It is common in the top-down literature to refer to 'correction' and correct (verb) as the change from the bottom-up prior, but this does not mean that the top-down results are correct. We have changed the language and added additional discussion of the uncertainty of top-down methods throughout the text per your suggestions.

Lines 118-119: state that satellite observations are distributed throughout the year, but are most dense during the southern hemisphere dry season (June-September). How the lower density of observations during the wet season in comparison with the dry season could affect the posterior estimates. This is particularly relevant given that this region has extensive wetland areas, where the highest emissions are expected during the wet season.
Thank you for this comment. Since we have higher observation density in the dry season (June-September), our emissions estimates are more heavily based on observations during this period. We have now included a figure in the supplement showing seasonal TROPOMI and GOSAT observation density (Figure S1). We also show in Figure S1 that the mean bias between GEOS-Chem and TROPOMI+GOSAT XCH$_4$ is lower in the posterior than in the prior in all seasons.

Figure 1 illustrates the annual mean 2021 dry-column methane mixing ratios (XCH4) after subtraction of background, clearly demonstrating the absence of TROPOMI data in the Amazon region, which is compensated by GOSAT observations. However, an examination of the plot of the number of observations shows that there are fewer GOSAT observations during the months of April to June in comparison with other months. Please describe the extent of data coverage for South America during this period, with particular attention to the Amazon region. It would be beneficial to have a map as supplementary material that includes both TROPOMI and GOSAT dry-column methane mixing ratios (XCH4) for the initial period of the year, particularly April to June.
Thank you. As you suggest, we have now included a figure in the supplement showing monthly TROPOMI and GOSAT observation density (Figure S1).

Lines 196-200: What is the lifetime of methane considering all the sinks, including oxidation by hydroxyl (OH) radicals and tropospheric chlorine (Cl), oxidation in the stratosphere, and uptake by soils?
Thank you for this question. The lifetime is $9.1 \pm 0.9$ years (Szopa et al., 2021), which we state on line 45 of the manuscript. We have also added this on line 227 for clarity.

Line 328: "Most of that increase is from anthropogenic emissions". Does this imply that the prior estimated wetland emissions for the South American region are consistent with the atmospheric

measurements? Alternatively, could the posterior wetland fluxes be more dependent on the prior estimates due to the limited observations in the Amazon region (which has larger methane emissions, as illustrated in Figure 2), as reflected in the low averaging kernel sensitivities? It would be beneficial to conduct a comparison with independent atmospheric observations to evaluate the posterior estimates.

That sentence does not imply that the prior wetland emissions are consistent with the satellite observations, but rather that the upward adjustment to wetland emissions is smaller than the total upward adjustment to anthropogenic emissions. This is a great point that this could be dependent on the lower averaging kernel sensitivities. In section 3.2, we describe the adjustments to wetland emissions in greater detail. Despite limited observations over the Amazon, we obtain an averaging kernel sensitivity of 0.74 over the region, indicating that our estimate is more informed by the observations than by the prior estimate.